# A rich conformational palette underlies human Ca_V2.1-channel availability

Kaiqian Wang[1], Michelle Nilsson [1], Marina Angelini [2], Riccardo Olcese [2,3], Fredrik Elinder [1,4] & Antonios Pantazis [1,5] ✉

Depolarization-evoked opening of Ca_V2.1 (P/Q-type) Ca$^{2+}$-channels triggers neurotransmitter release, while voltage-dependent inactivation (VDI) limits channel availability to open, contributing to synaptic plasticity. The mechanism of Ca_V2.1 response to voltage is unclear. Using voltage-clamp fluorometry and kinetic modeling, we optically track and physically characterize the structural dynamics of the four Ca_V2.1 voltage-sensor domains (VSDs). The VSDs are differentially sensitive to voltage changes, both brief and long-lived. VSD-I seems to directly drive opening and convert between two modes of function, associated with VDI. VSD-II is apparently voltage-insensitive. VSD-III and VSD-IV sense more negative voltages and undergo voltage-dependent conversion uncorrelated with VDI. Auxiliary β-subunits regulate VSD-I-to-pore coupling and VSD conversion kinetics. Hence, the central role of Ca_V2.1 channels in synaptic release, and their contribution to plasticity, memory formation and learning, can arise from the voltage-dependent conformational changes of VSD-I.

The Ca_V2.1, or P/Q-type, voltage-gated Ca$^{2+}$ channel, is the predominant Ca_V subtype in the brain, and it plays a crucial role in synaptic transmission[1–8]. Presynaptic Ca_V2.1 channels convert an electrical signal (action potentials) into a biochemical signal (Ca$^{2+}$ entry), triggering neurotransmitter release (Fig. 1a). A prolonged depolarization or train of action potentials cause Ca_V2.1 voltage-dependent inactivation (VDI)[9,10]. During VDI, channels enter a non-conductive state and are not available to mediate Ca$^{2+}$ influx. This contributes to short-term depression, a form of synaptic plasticity that affects informational encoding[11–14] (Fig. 1b). Ca_V2.1 channels can also undergo voltage-dependent facilitation, which depends on direct channel regulation mediated by neurotransmitters' action on GPCR and activating G-proteins[9]. Postsynaptic Ca_V2.1 channels generate depolarization-induced local Ca$^{2+}$ transients and are implicated in long-term depression, which underlies cerebellar learning[4,15]. Mutational studies in mice suggest a role of Ca_V2.1 in synaptic plasticity, spatial learning, and memory; while variants of *CACNA1A*, the gene encoding the Ca_V2.1 pore-forming subunit α_{1A}, are associated with serious neurological disease[16–18].

Ca_V2.1 channels consist of the transmembrane α_{1A}-subunit, extracellular α_2δ-subunit, and intracellular β-subunits[7,8]. Channel voltage regulation stems from voltage-dependent conformational changes in the α_{1A} voltage-sensing apparatus[19,20]. This comprises four transmembrane, homologous but non-identical, voltage-sensor domains (VSDs; Fig. 1c–e), but their roles in voltage-dependent activation and inactivation were not previously studied. Because Ca_V2.1 has four different VSDs, it is possible that each VSD serves different functions to drive neurosecretion and contribute to synaptic plasticity. We have reported such functional heterogeneity among the VSDs of L-type (Ca_V1.1, Ca_V1.2)[21–23] and N-type (Ca_V2.2)[24] channels.

Here, we optically track the voltage-dependent movements of the individual Ca_V2.1 VSDs under physiologically-relevant conditions by combining the cut-open oocyte vaseline gap voltage clamp[25–27] with voltage-clamp fluorometry (VCF)[27–31]. By "physiologically relevant

[1]Division of Cell and Neurobiology, Department of Biomedical and Clinical Sciences, Linköping University, Linköping, Sweden. [2]Department of Anesthesiology and Perioperative Medicine, David Geffen School of Medicine, University of California, Los Angeles, CA, USA. [3]Department of Physiology, David Geffen School of Medicine, University of California, Los Angeles, CA, USA. [4]Science for Life Laboratory, Linköping University, Linköping, Sweden. [5]Wallenberg Center for Molecular Medicine, Linköping Univincity, Linköping, Sweden. ✉ e-mail: antonios.pantazis@liu.se

conditions", we mean an *in-cellula* study of the human, conducting, $Ca_V2.1$ pore-forming isoform, reconstituted with essential auxiliary subunits. We find that each VSD has a different response to brief depolarizations and long-term changes in the holding potential, which is tuned by $Ca_V\beta$ auxiliary subunits. This predicts channel populations with a highly diversified set of VSD conformations, especially near resting voltages relevant to neuronal physiology. The stark exception is VSD-II, which appears to be unresponsive to voltage changes. Our results suggest that activation of the first $Ca_V2.1$ voltage-sensor domain (VSD-I) drives excitation-evoked presynaptic $Ca^{2+}$ influx. In addition, conversion of VSD-I into a form not compatible with channel opening determines channel availability to conduct and thus underpins VDI—a form of molecular memory that contributes to memory formation and learning in the brain.

## Results

### $Ca_V2.1$ VSDs activate with distinct voltage dependencies

To optically track the movements of individual VSDs under physiologically relevant conditions, we used VCF. Briefly, specific amino-acid residues at the extracellular loop between the S3 and S4 transmembrane helices of each repeat were mutated to cysteine (Fig. 1e). During the experiments, the engineered cysteine was modified with the thiol-reactive and environment-sensitive fluorophore, MTS-TAMRA. Thus the conformational rearrangements of the labeled VSDs in response to brief depolarizations were reported as ensemble fluorescence deflections ($\Delta F$). To limit additional regulation ($Ca^{2+}$ regulation or VDI), we (i) used $Ba^{2+}$ as charge carrier, and pre-injected cells with the BAPTA $Ca^{2+}$-chelator[21]; and (ii) studied $Ca_V2.1$ channels including $\beta_{2a}$, which slows down VDI relative to β-less channels[32,33].

VSD-I activated with a two-part voltage dependence (Fig. 2a, f): one component, F1, had a voltage-dependence very close to that of pore opening (calculated by normalized tail current, $I_{tail}$), and the other (F2) was observed at very negative potentials (Table 1).

We only detected faint $\Delta F/F$ signals (<0.1%) from VSD-II (Fig. 2b), similar to signals from $Ca_V2.1$ without a substituted cysteine (Fig. 2e), likely due to non-specific labeling. Lack of $\Delta F$ suggested that VSD-II does not undergo voltage-dependent conformational changes. In fact, no $\Delta F$ were detected despite (i) probing most of the S3-S4 linker, (ii) trying different fluorophores, (iii) removing a tryptophan that might quench the nearby fluorophore[34–37]; (iv) using a different complement of auxiliary subunits; (v) neutralizing counter-charges that could stabilize the $S4_{II}$ resting state[19] and (vi) perturbing a $PIP_2$-binding site, resolved in a $Ca_V2.1$ structure[38] (Fig. S1). Indeed, the numerous VSD-II mutations tested did not result in a consistent or substantial alteration of the voltage-dependence of pore opening (Fig. S2), suggesting that VSD-II does not contribute to $Ca_V2.1$ voltage sensitivity.

VSD-III and VSD-IV appeared to activate at negative potentials, close to the physiological resting membrane potential ($V_{rest}$, Fig. 2c, d, g, h). As in our VCF investigations of $Ca_V1.1$ and $Ca_V2.2$ channels[23,24], the $\Delta F$ signals from VSD-IV had opposite sign to those resolved from VSD-I and VSD-III (Fig. 2d). A straightforward interpretation is that, when MTS-TAMRA labels VSD-I and VSD-III, it is relatively more quenched in the active state than in the resting state; and vice-versa for VSD-IV. The voltage-dependence of VSD-IV appeared shallower than VSD-III, suggesting that VSD-IV is less sensitive to voltage changes. VCF mutations and labeling did not substantially change the voltage dependence of pore opening (Fig. 2f–h). Figure 2i illustrates the diverse conformational responses of different parts of $\alpha_{1A}$ to brief depolarizations.

### Progressive VSD conversion under VDI-favouring conditions

$Ca_V2.1$ availability is limited by VDI[13] (Fig. 1a, b), here recapitulated by changing the holding membrane potential ($V_h$; Fig. 3a). Which VSD is responsible for VDI? The VSD-I two-part response to membrane depolarization (Fig. 2f) suggested the presence of two $Ca_V2.1$

populations: one whose VSD-I activated with similar voltage-dependence to pore opening and another whose VSD-I activated at very negative potentials, far (along the voltage axis) from pore opening. The latter process was reminiscent of charge interconversion: the observations that charge movement (i.e., the overall activation of all VSDs measured by gating currents) occurs at more negative potentials as $Ca_V$ channels enter inactivated states during prolonged depolarization[39,40]. At negative $V_h$ (−80 mV), a brief pulse to 40 mV produced robust VSD-I activation (Fig. 3b). In contrast, no VSD-I movements were detectable using the same step when $V_h$ was very positive (40 mV, Fig. 3c), when channels were inactivated. In the presence of VDI-accelerating $\beta_3$-subunits[32,33], fewer VSD-I could activate at $V_h$ = −80 mV, compared with $\beta_{2a}$ (Fig. 3b, d), and no movements were detected at $V_h$ = 40 mV (Fig. 3e). These observations hinted that VSD-I is linked to VDI.

To explore $V_h$-dependent VSD-I conversion in detail we used a broad range of $V_h$ and brief (100-ms) test potentials ($V_t$). Upon more positive $V_h$, the proportion of channels with VSD-I with depolarized voltage-dependence (F1) progressively diminished (Fig. 3f; Boltzmann parameters in Table S1), converting to channels whose VSD-I activated at hyperpolarized potentials (F2). Because of the large separation between F1 and F2 (120 mV along the voltage axis), the conversion can be said to drastically alter the biophysical properties of VSD-I. Plots of the first derivatives of the voltage-dependence curves (Fig. 3g) and F1 percentage versus $V_h$ (Fig. 3h) better illustrate F1-F2 interconversion, which occurred around $V_{rest}$. In $\beta_3$-containing channels, conversion to F2 was favored, occurring at more negative $V_h$ (Fig. 3i–k).

VSD-III and VSD-IV also converted (Fig. 4, Table S1) and their apparently one-part voltage-dependences (Fig. 2g,h) could be reinterpreted as mixtures of two populations. The gap between F1 and F2 VSD-III activations was approximately half as wide as that of VSD-I ($\sim$ 60 mV), suggesting that VSD-III is less altered by conversion. The F1 and F2 components of VSD-IV had strikingly different apparent voltage sensitivity. $\beta_3$ facilitated both VSD-III and VSD-IV conversion.

Summarizing our findings so far, the $Ca_V2.1$ VSDs exhibit diverse responses to both transient depolarization (Fig. 2) and changes in the holding potential (Figs.3,4).

### VSD-I conversion is linked to inactivation

Fitting fluorescence data to the sum of two Boltzmann functions provided a good empirical overview, but it had two shortcomings: (i) it implied that channel transitions within F1 and F2 are independent of each-other, and (ii) it did not account for kinetics. To characterize VSD activation and conversion with more mechanistic rigor, VCF data from each VSD were fit to a four-state model (Fig. 5a). The kinetic model combines VSD activation and deactivation (responses to brief potential changes) with interconversion between two modes of gating (responses to holding potential changes). Mode 1 corresponds to the F1 component from the Boltzmann-distribution fits, and mode 2 to F2. However, the four-state model is physically more meaningful, accounting for the kinetics of all transitions while obeying microscopic reversibility, which implies charge conservation. The model fit to the data is shown in Fig. S3, the optimized parameters in Table S2, and the calculated equilibrium constants (the quotients between the forward and backward rates) in Table S3.

VSD-I converted from mode 1 to mode 2 spontaneously, i.e., in a voltage-independent manner. Conversion occurred preferentially from state A1 ($k_{con}$ = 0.77 s$^{-1}$), while recovery occurred from state R2 ($k_{rec}$ = 0.16 s$^{-1}$). Since A1 is visited at depolarized potentials, and R2 at hyperpolarized potentials, the distribution of channels in mode 1 or mode 2 had an apparent voltage dependence, with a $V_{0.5} \cong$ −60 mV (Fig. 5b). VSD-III and VSD-IV also converted spontaneously from the active state but, in contrast to VSD-I, the data were not consistent with a voltage-independent transition between R1 and R2. Instead, an intrinsically-voltage-dependent transition was required ($z_{R1\leftrightarrow R2}$ = 1.0

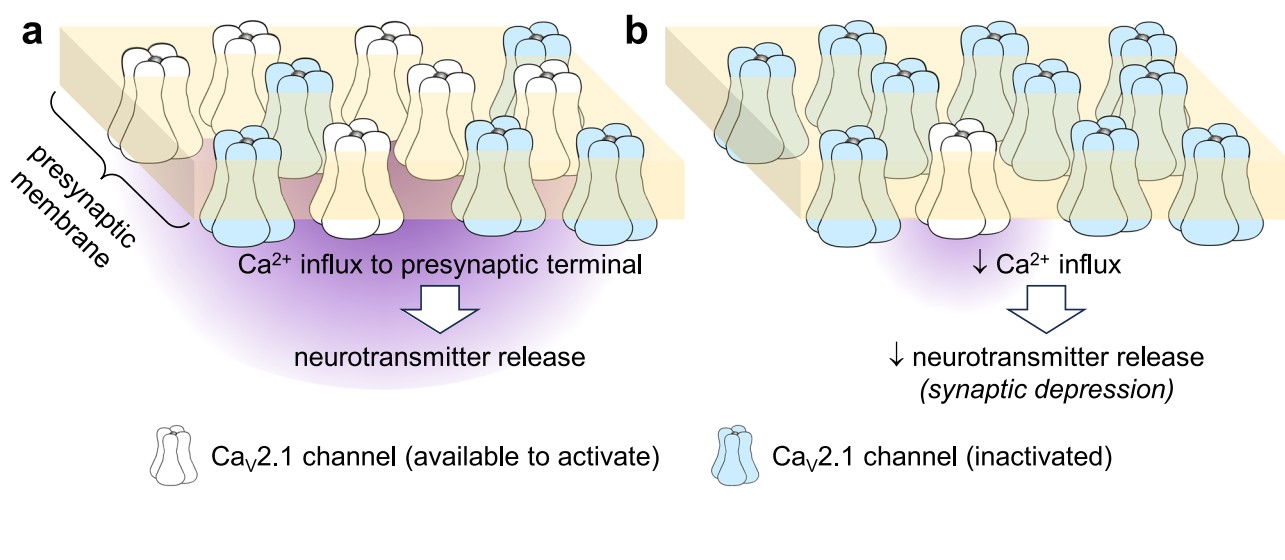

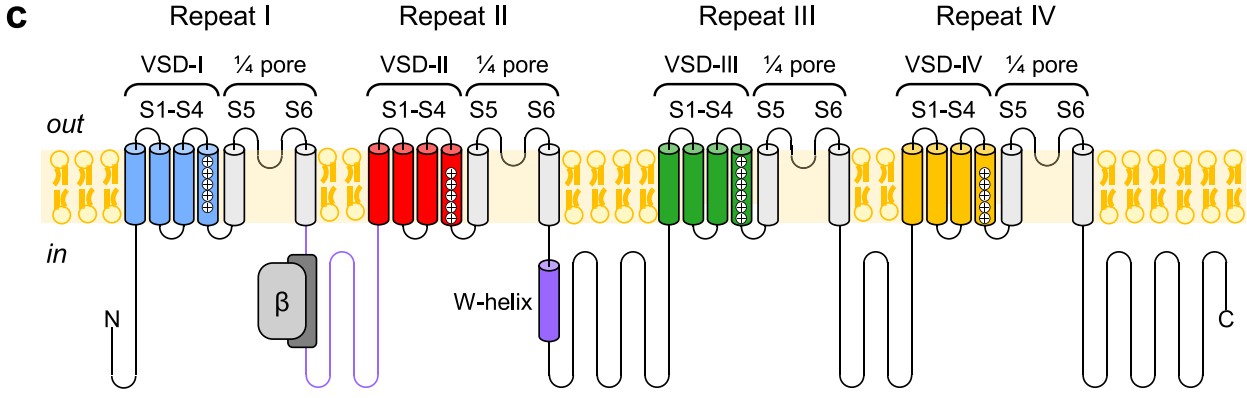

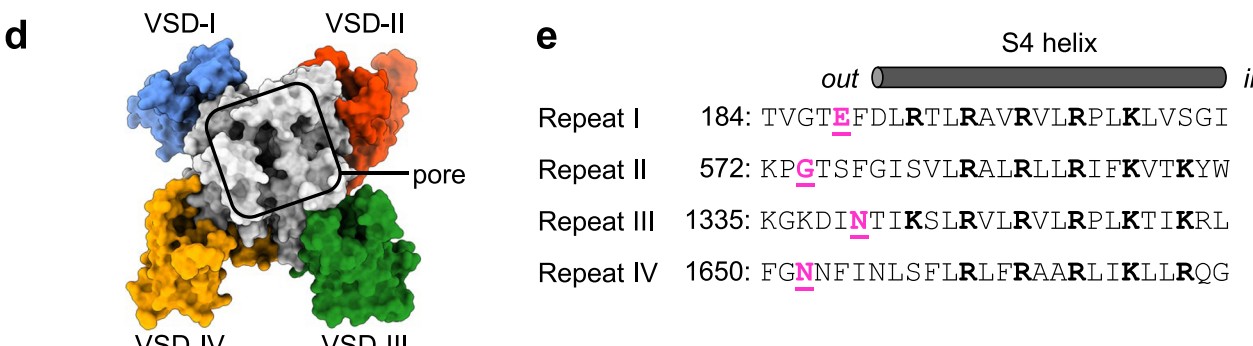

**Fig. 1 | Ca$_V$2.1 function and consequences of inactivation, its pore-forming subunit, and its four non-identical VSDs. a** Under normal conditions, some presynaptic Ca$_V$2.1 channels are available to activate (white) in response to an action potential, mediating Ca$^{2+}$ influx into the presynaptic terminus that triggers transmitter release. Some Ca$_V$2.1 channels are inactivated (blue). **b** Prolonged depolarization or trains of action potentials induce voltage-dependent inactivation (VDI), further decreasing the number of available Ca$_V$2.1 channels and subsequently transmitter release. This contributes to synaptic plasticity. **c** The Ca$_V$2.1 pore-forming subunit ($\alpha_{1A}$) contains four homologous repeats (I-IV). Membrane-spanning helices S1-S4 from each repeat comprise a voltage-sensor domain (VSD). The S5-S6 helices from each repeat form the ion-conducting pore. The auxiliary β-subunit binds between repeats I and II[53]. The intracellular I-II linker and W-helix within the II-III linker (indigo) act as blocking particles to occlude ion conductance during VDI in related Ca$_V$2 channels[45–48]. **d** Top view of $\alpha_{1A}$ (PDB: 8X90[38]). **e** S4 helix sequence comparison. Positively charged residues (bold) confer voltage sensitivity to the VSDs[19,20]. Amino-acid residues substituted to cysteine for fluorescence labeling in Fig. 2 are in magenta: VSD-I: E188; VSD-II: G574; VSD-III: N1340; VSD-IV: N1652.

and 0.65 $e_0$, respectively; Table S2). Their steady-state modal inter-conversion occurred at more negative potentials (ca. −80 mV Fig. 5b, Table S2) than for VSD-I.

Changing from $\beta_{2a}$- to $\beta_3$-subunits altered several biophysical parameters. Of note: First, the voltage dependence of VSD-I activation in mode 1 was shifted to more negative voltages, separating from pore opening by ca. 25 mV (Table S2). The activation transitions of other VSDs, and all activation transitions in mode 2, were relatively less affected. Second, the VSD-I conversion equilibrium constant from the active state increased by 8-fold, which resulted in a shift of

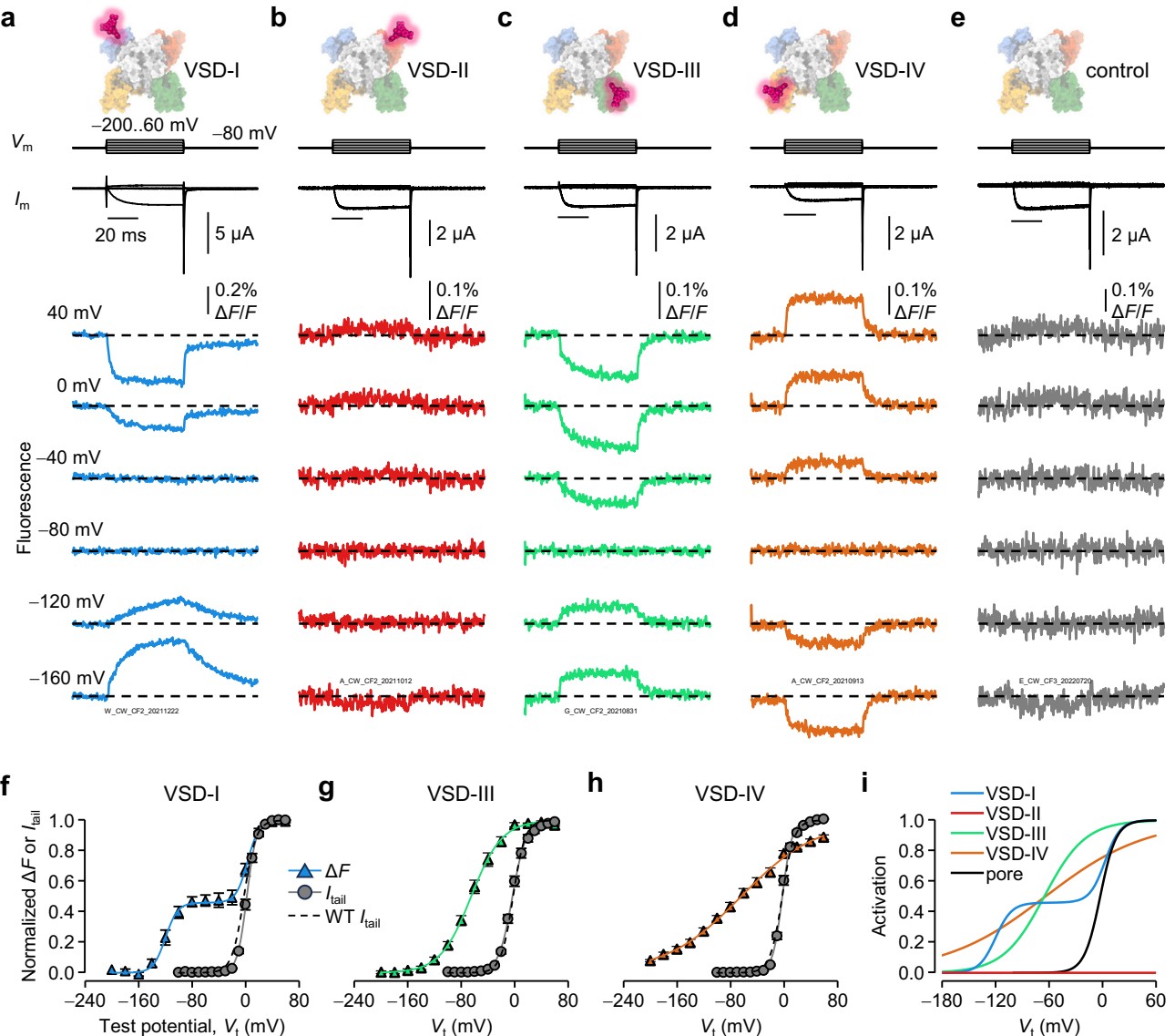

**Fig. 2 | Ca$_V$2.1 VSDs have diverse voltage-dependent activation properties.**
**a** VCF recordings of Ca$_V$2.1 complexes ($\alpha_{1A}/\alpha_2\delta-1/\beta_{2a}$) fluorescently labeled in VSD-I. Voltage steps ($V_m$) are shown on top; ionic currents ($I_m$) and fluorescence deflections ($\Delta F$) were acquired simultaneously. **b** As in (**a**) for VSD-II. VSD-II does not show clear $\Delta F$ and appears to be voltage-insensitive (Figs. S1, S2). **c**, **d** As in (**a**) for VSD-III and VSD-IV, respectively. **e** As in (**a**) for control channels (no substituted Cys). **f** Voltage dependence of VSD-I activation (normalized $\Delta F$, blue triangles) and fit to the sum of two Boltzmann distributions (blue curve, Eq. 3). Voltage

dependence of pore opening (normalized $I_{tail}$, Eq. 1) for VSD-I-labeled channels (gray circles and curve). The voltage dependence of pore opening for control channels is shown as black dashed curve. All voltage-dependence parameters are in Table 1. **g** As in (**f**), for VSD-III (green; Eq. 2). **h** As in (**f**), for VSD-IV (orange). **i** Overlay of all voltage dependences observed on the human $\alpha_{1A}$ subunit. Blue: VSD-I; green: VSD-II; orange: VSD-IV; black: pore opening (from wild-type channels). VSD-II activation is shown as a flat red line. Error bars are S.E.M.

the steady-state conversion voltage-dependence by −25 mV. Likewise, the conversions of VSD-III and VSD-IV were facilitated, resulting in similar (but smaller) negative shifts (Fig. 5b, c, Tables S2 & S3).

Most pertinent to VDI, the fraction of channels with a VSD-I in mode 1 and the fraction of channels available to activate (i.e., non-inactivated), were statistically indistinguishable. By contrast, the fraction of channels with VSD-III or VSD-IV in mode 1 was statistically distinct from the fraction of non-inactivated channels (Fig. 5d).

## Discussion

We have experimentally and analytically shown that the four Ca$_V$2.1 VSDs display distinct conformational changes. VSDs undergo activation transitions over a very broad range of voltages (Fig. 2i). A study using gating-current measurements reported charge movement (i.e., VSD transitions) at only depolarized voltages[41]; we believe a reason for

this discrepancy is that the capacitance compensation and P/N subtraction protocols used to measure gating currents likely hindered the detection of charge movement at negative voltages.

The VSDs do not merely possess quantitatively distinct biophysical properties, but exhibit qualitative differences in their structural dynamics. VSD-I movements closely correlate with channel opening and VDI (Figs. 2f, 5d). VSD-II appears to be voltage-insensitive (Figs. 2, S1, S2). VSD-III and VSD-IV exhibit a voltage-dependent conversion between the resting states (Table S2), which results in a steady-state occupancy of the mode-2 resting state (R2) over physiological $V_{rest}$ (Fig. 6a, b): a unique feature, as R2 is a metastable state in canonically-converting VSDs[42,43], like VSD-I. To better illustrate the multiplicity of VSD steady-state conformations, we mapped the state occupancies of each VSD into state spectra (Fig. 6a, b), used to color the Ca$_V$2.1 structure from extremely hyperpolarized voltages, through

**Table 1 | Voltage-dependence parameters of $Ca_V2.1$ pore opening and VSD activation**

|  | WT | VSD-I |  | VSD-III |  | VSD-IV |  |
|---|---|---|---|---|---|---|---|
|  | $I_{tail}(V)$ | $I_{tail}(V)$ | $\Delta F(V)$ | $I_{tail}(V)$ | $\Delta F(V)$ | $I_{tail}(V)$ | $\Delta F(V)$ |
| $V_{0.5,F1}$ (mV) | −2.2 ± 0.86 | 2.0 ± 1.1 | 3.1 ± 1.0 | −3.0 ± 1.4 | −65 ± 3.8 | −2.0 ± 1.0 | −64 ± 6.3 |
| $z_{F1}$ ($e_0$) | 2.9 ± 0.091 | 3.6 ± 0.11 | 3.0 ± 0.39 | 2.5 ± 0.16 | 1.1 ± 0.053 | 3.5 ± 0.15 | 0.45 ± 0.017 |
| $F1$ (%) | N/A | N/A | 54 ± 3.9 | N/A | 100 | N/A | 100 |
| $V_{0.5,F2}$ (mV) | N/A | N/A | −120 ± 2.7 | N/A | N/A | N/A | N/A |
| $z_{F2}$ ($e_0$) | N/A | N/A | 2.8 ± 0.32 | N/A | N/A | N/A | N/A |
| $n$ (cells) | 12 | 4 | 4 | 7 | 7 | 6 | 6 |

Representative traces and curves in Fig. 2. Errors are S.E.M.

voltages pertinent to physiological $V_{rest}$, to the fully depolarized membrane (Fig. 6c, Supplementary Movie 1).

Depolarization promotes VSD activation (Fig. 2). While VSD-III and VSD-IV are fastest to activate, a major finding is that VSD-I activation (in mode 1) and pore opening occur over the same membrane potential (Fig. 2, Table S2). We propose that such processes may be called "syntasic", from classical Greek syn (σύν, together) and tasis (τάσις, tension, or in this case, voltage). The strong connection between VSD-I activation to A1 and pore opening suggests that the former is the first molecular transition that triggers neurotransmitter release in most synapses.

Figure 6 shows that $Ca_V2.1$ channels exhibit the most conformational diversity around $V_{rest}$. At these voltages, $Ca_V2.1$ channels exist with VSDs either resting in mode 1, active in mode 2, or (in the case of VSD-III and VSD-IV) resting in mode 2; and channels can be closed or inactivated. Yet, certain combinations of conformations are more favored (Fig. 5d): essentially, $V_{rest}$ bisects the $Ca_V2.1$ population into channels with VSD-I in mode 1, primed to trigger neurosecretion; and channels with VSD-I in mode 2 and inactivated, but available to be recruited when $V_{rest}$ becomes more negative.

Since $Ca_V2.1$ VDI contributes to synaptic plasticity mechanisms[13,14,17,18], another VSD-I transition−in this case conversion− is shown to be linked to processes of a scale well beyond intramolecular structural dynamics: cognition, and memory formation. In addition, presynaptic $Ca_V2.1$ availability may serve to stabilize synaptic release, similar to the stabilization of firing in axons by sodium-channel slow inactivation[44]. A straightforward mechanism for how VSD-I conversion triggers inactivation is that, since VSD-I A1 is linked to pore opening, inability to achieve A1 would produce channels unavailable to conduct. VSD-I conversion may, as a conformational change, also play an active role in VDI development, engaging cytosolic structures lacking intrinsic voltage dependence yet associated with inactivation, such as the hinged lid[45] and the W-helix[46–48].

Every single spike carries a small probability of pushing more VSD-Is to the converted (mode-2) states: the onset of VSD-I conversion is a rare event for any single $Ca_V2.1$ channel ($k_{con}$ = 0.77 or 1.4 s⁻¹ with $\beta_{2a}$ or $\beta_3$, respectively; Table S2). This translates to a population transition with kinetics of about one second. Recovery from VSD-I mode 2 is about five times slower ($k_{rec}$ = 0.16 or 0.22 s⁻¹ with $\beta_{2a}$ or $\beta_3$, respectively; Table S2). Compared to the millisecond kinetics of a neuronal action potential, VSD-I recovery from conversion is a far slower event, so one could describe conversion as a form of molecular memory. And yet, as a process that contributes to acquiring lifelong memories, VSD-I conversion is also one of the fastest events on the brain-wide time-scales of learning[49].

VSD-II appears to consistently lack voltage-dependence, both in $Ca_V2.1$ (Figs. 2, S1, S2) and $Ca_V2.2$[24]. An inability to undergo voltage-dependent movements explains why, in all $Ca_V2$-channel structures reported, VSD-II was resolved in a resting conformation[38,46–48,50] despite the absence of an electric field (equivalent to $V_h$ = 0 mV). By the same token, VSD-II being locked down in the resolved $Ca_V2$ structures supports the lack of optical signals reported here and for $Ca_V2.2$[24]. In the

structures, all other VSDs were resolved in an active state; this is in agreement with our work, which predicts that the majority of VSD-I, VSD-III, and VSD-IV would be in a mode-2 active state at $V_h$ = 0 mV (Fig. 6). Whether VSD-II can activate under different conditions (such as the signaling milieu or additional protein partners) is an outstanding question. Another consideration is that the lipidic composition of the plasma membrane in our heterologous expression system (*Xenopus laevis* oocytes) differs from that in human neurons[51,52].

Both VSD-III and VSD-IV activate faster than VSD-I and at more negative potentials (Table S2); they could possess regulatory roles in the activation process. In $\beta_3$-containing channels, where VSD-I activation and pore opening are "asyntasic" ($V_{0.5}$ of −22 and 5 mV, respectively; Table S2), voltage-dependent opening could be a more cooperative process involving VSD-III and VSD-IV. A plasticity of VSD-pore connectivity following auxiliary-subunit changes has been reported in $Ca_V1.2$[22]. A common feature of $Ca_V$-channel VSDs is that, despite their homology, their non-identity translates to functional heterogeneity−there are functional differences both within and between different $Ca_V$ isoforms[21,23,24].

$\beta_{2a}$, relative to $\beta_3$, strongly inhibited conversion by shifting the overall steady-state conversion to more positive potentials (Figs. 3–5, Table S2). VSD-I and VSD-III are more affected by changes in $\beta$-subunit composition than VSD-IV. The most pronounced effects of $\beta_{2a}$ are diminished conversion equilibrium constants from the active state by more than 6-fold in VSD-I and VSD-III (by comparing rates in table S2) and also diminished the conversion equilibrium constant between the resting states by ~6-fold in VSD-III (at −80 mV; calculated from parameter values in table S2 using Eq. 8). $\beta$-subunits bind to the cytosolic I-II loop[45,53] (Fig. 1a). While we cannot exclude allosteric effects of $\beta$-subunits on VSD structural dynamics, $\beta$-subunits may also interact directly with the cytosolic VSD flanks, in a state-dependent manner, similar to the proposed action of G-proteins on $Ca_V2.2$[24]. While both $\beta_2$ and $\beta_3$ subunits are expressed in the brain, a highly abundant $Ca_V\beta$ isoform is $\beta_4$[6]. Since the initial rate of inactivation in the presence of $\beta_4$ is roughly half-way between $\beta_{2a}$ (slowest) and $\beta_3$ (fastest)[32], we anticipate that the properties of native $Ca_V2.1$ VSDs are in between those reported here.

Our work here has uncovered a particularly rich gamut of conformations of the $Ca_V2.1$ VSDs, as they respond to electrical signals both transient and long-lived, tuned to both the resting membrane potential and depolarization, and under the influence of different $\beta$-subunits. Yet this is only a part of the regulation $Ca_V2.1$ is subject to in their presynaptic environment: several molecular partners, including $G\beta\gamma$[54,55], calmodulin[56], and CaBP1[57], as well as neuronal junctophilins[58] and syntaxin[59], can modulate $Ca_V2.1$ voltage-dependent activation and inactivation. It will be of high interest to investigate whether they act via the same pathway as $\beta$-subunits or whether the $Ca_V2.1$ voltage-sensing apparatus possesses specific handles for each regulatory partner. Given the importance of $Ca^{2+}$-signal amplitude and timing for synaptic communication, it is fitting that its principal mediator is a macromolecule with exquisite structural dynamics and regulation.

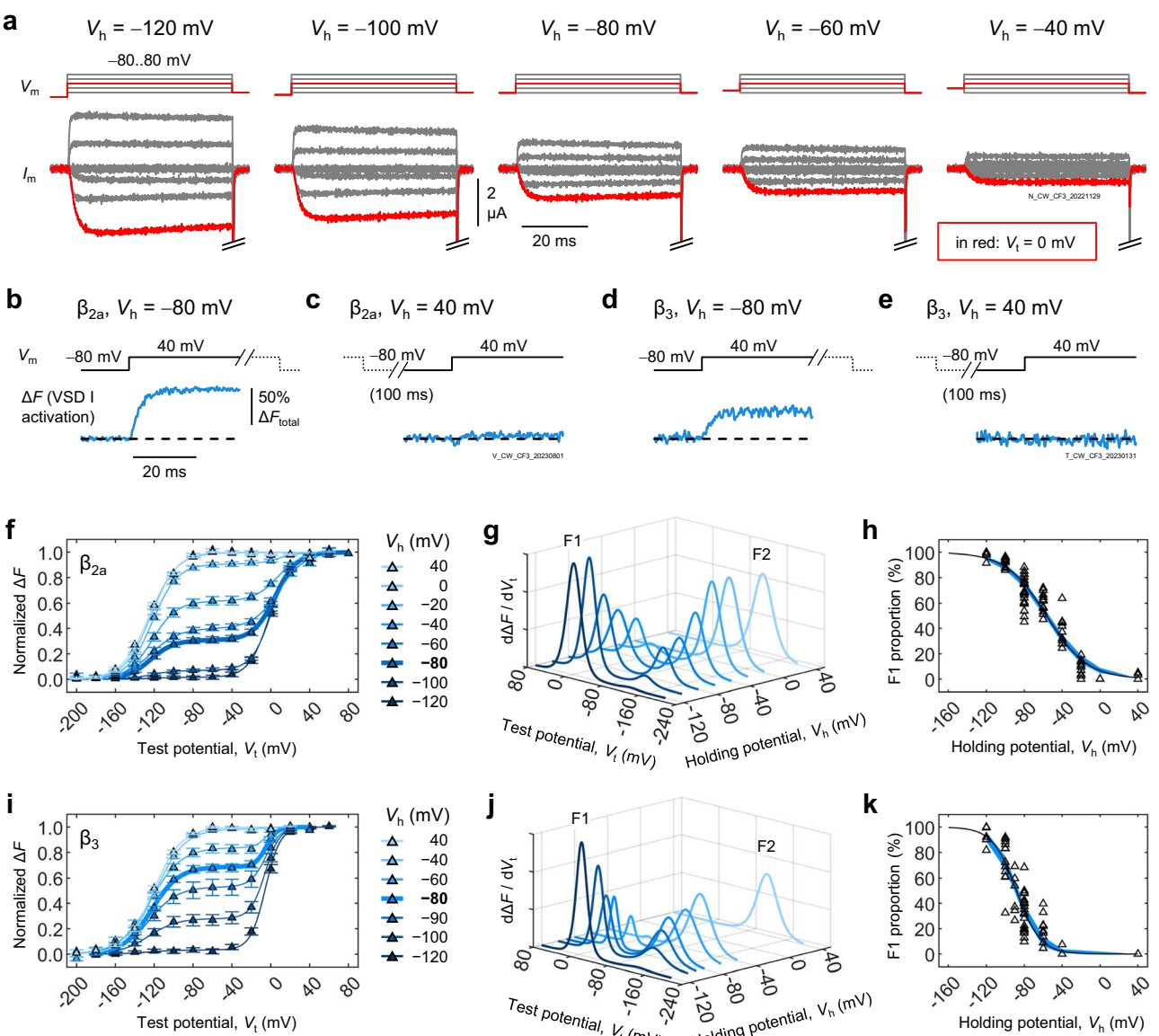

**Fig. 3 | VSD-I converts under VDI-favouring conditions. a** Voltage steps ($V_m$) and exemplary currents ($I_m$) from a cell expressing $Ca_V2.1$ channels ($\alpha_{1A}/\alpha_2\delta$-1/$\beta_{2a}$) at different holding potentials ($V_h$). Tail currents were cropped out for clarity. The current (i.e., channel availability) decreased as $V_h$ became more positive: the hallmark of VDI[13]. **b–e** VSD-I activation (blue traces) in response to the same voltage step (black; −80 to 40 mV) under different VDI regimes: (**b**) $\beta_{2a}$ subunits, $V_h = -80$ mV (VDI low); (**c**) $\beta_{2a}$, $V_h = 40$ mV (VDI high); (**d**) $\beta_3$, $V_h = -80$ mV (VDI intermediate); (**e**) $\beta_3$, $V_h = 40$ mV (VDI high). The −80-mV steps in (**c**, **e**) were 100 ms long. **f** Voltage dependence of VSD-I activation at different $V_h$ in the presence of $\beta_{2a}$. Solid curves are the sums of two Boltzmann distributions (Eq. 3; parameters in

Table S1). The lightness of symbols and curves increases as $V_h$ becomes more positive. Error bars are S.E.M. **g** The first derivatives of the curves from (**f**) illustrate the conversion of VSD-I from F1 to F2 as $V_h$ becomes more positive. **h** Apparent voltage dependence of VSD-I conversion. Open triangles are individual data; the blue surface is the 95% confidence interval of a Boltzmann fit (Eq. 4; $V_{0.5} = -56.4$ [−59.0, −53.9] mV; $z = 1.18$ [1.05, 1.31] $e_0$, $n = 43$ cells). **i–k** As in (**f–h**), respectively, for channels with $\beta_3$. F1-F2 conversion occurs at more negative voltages: ($V_{0.5} = -88.2$ [−90.4, −86.0] mV; $z = 2.00$ [1.62, 2.37] $e_0$, $n = 23$). Parameter values given as the mean and 95% confidence interval [lower bound, upper bound].

## Methods

### Ethical statement

All animal experiments were approved by the Linköping University Animal Care and Use Committee (document number 15839-2018, protocol number 1941).

### Molecular biology

The human *CACNA1A* transcript variant 3 (EFa, NM_001127221.2, Uniprot O00555.3) was codon-optimized for *Xenopus laevis* expression by Integrated DNA Technologies (IDT) and subcloned into the Z-vector[60]. All site-directed mutagenesis was performed with a high-fidelity *Pfu* polymerase (Agilent 600850) and confirmed by full-gene DNA

sequencing. Molecular biology reagents were obtained from New England Biolabs, and synthetic oligonucleotides from IDT. In vitro cRNA transcription was performed with the AmpliCap-Max T7 High Yield Message Maker Kit (Cellscript); RNA was stored at −80 °C in RNA storage solution (Thermo Fisher Scientific). $\alpha_{1A}$ subunits were coexpressed with rabbit $\alpha_2\delta$-1 (*CACNA2D1*, Uniprot P13806) and either rat $\beta_{2a}$ (*CACNB2A*, UniProt Q8VGC3) or rabbit $\beta_3$ (*CACNB3*, Uniprot P54286) subunits.

### Oocyte preparation and labeling

Defolliculated *Xenopus laevis* (Nasco) oocytes (stage V-VI) were purchased from Ecocyte (grade I, defolliculated) or prepared as follows.

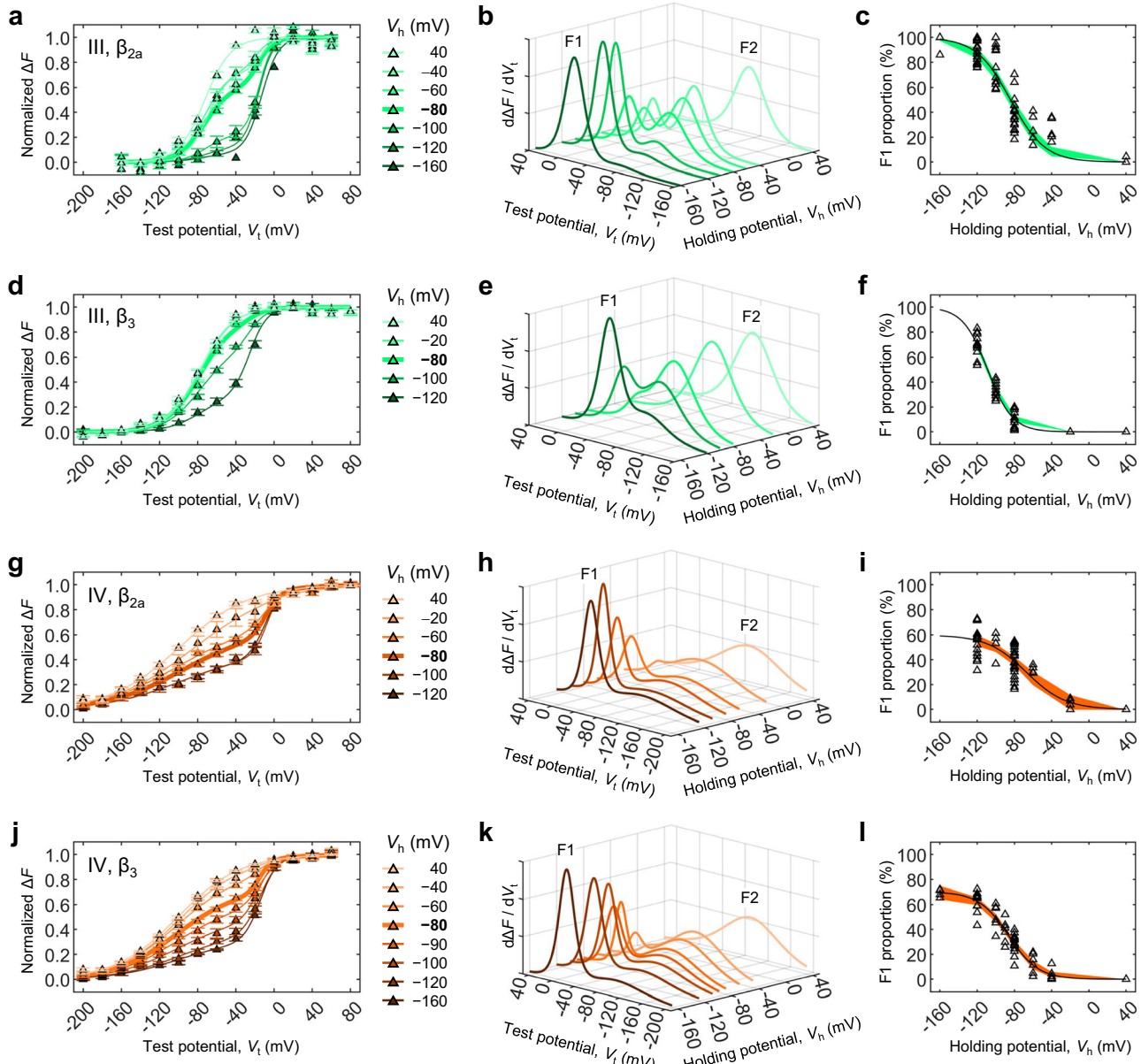

**Fig. 4 | VSD-III and VSD-IV can convert.** VSD-III and VSD-IV activation at extended holding and test potentials ($V_h$, $V_t$) revealed that they also undergo conversion. Similar to VSD-I, the voltage dependence of VSD-III and VSD-IV at $V_h = -80$ mV (Fig. 2g, h) consisted of transitions in a mixed population of F1 and F2. **a** Voltage dependence of VSD-III activation in the presence of $\beta_{2a}$. Solid curves are the sums of two Boltzmann distributions (Eq. 3, parameters in Table S1). Error bars are S.E.M. **b** The first derivatives of the curves from (a) illustrate the conversion of VSD-III from F1 to F2 as $V_h$ becomes more positive. **c** Apparent voltage-dependence of VSD-III conversion. Open triangles are individual data; the green surface is the 95% confidence interval of a Boltzmann fit (Eq. 4; $V_{0.5} = -84.2$ [−87.3, −81.1] mV; $z = 1.37$

[1.11,1.63] $e_0$, $n = 19$ cells). **d**–**f** As in (**a**–**c**), respectively, for channels complexed with $\beta_3$. The F1-F2 transition occurs at more negative voltages: ($V_{0.5} = -109$ [−110, −107] mV; $z = 1.91$ [1.69,2.12] $e_0$, $n = 13$ cells). **g**–**i** As in (**a**–**c**), respectively, for channels with $\beta_{2a}$ labeled in VSD-IV. In the Boltzmann fits of (**i**), the positive asymptote (F1$_{max}$) was a free parameter: F1$_{max} = 57.7$ [49.9,65.5] %; $V_{0.5} = -65.4$ [−73.6, −57.2] mV; $z = 1.28$ [0.658,1.91] $e_0$, $n = 26$ cells. **j**–**l** As in (**g**–**i**), respectively, for VSD-IV-labeled channels with $\beta_3$. F1$_{max} = 70.2$ [64.0,76.3] %; $V_{0.5} = -86.3$ [−90.5, −82.1] mV; $z = 1.58$ [1.20,1.97] $e_0$, $n = 15$ cells. Parameter values given as the mean and 95% confidence interval [lower bound, upper bound].

Oocyte lobes were surgically removed and separated into clusters of up to 5 oocytes. The follicular layer was removed (i) enzymatically with Liberase (Roche) and (ii) mechanically. Both steps were done using an orbital shaker at 88 rpm at room temperature in Ca$^{2+}$-free solution OR-2 (in mM: 82.5 NaCl, 2.5 KCl, 1 MgCl$_2$, 5 HEPES, pH = 7.0), first with Liberase for approximately 20 min, then in OR-2 for approximately 45-75 min. Cells were stored in SOS (in mM: 100 NaCl, 2 KCl, 1.8 CaCl$_2$, 1 MgCl, 5 HEPES, pH = 7.0) at 17 °C.

Each oocyte was micro-injected with a 50 nL cRNA mixture of $\alpha_{1A}$, $\alpha_2\delta$-1, and either $\beta_{2a}$ or $\beta_3$ (0.6-0.8 µg/µL of each subunit). Oocytes

were incubated at 17 °C in 0.5 × Leibovitz's L-15 (Corning) diluted in MilliQ H$_2$O, supplemented with 1% horse serum (Capricorn Scientific), 100 units/mL penicillin and 100 µg/mL streptomycin (Gibco), 100 µg/mL amikacin (Fisher BioReagents) for 4-6 days. Prior to fluorescence staining, oocytes were rinsed in SOS.

Oocytes expressing Cys-substituted Ca$_V$2.1 channel complexes were labeled with, unless otherwise stated, 20 µM MTS-5(6)-carboxytetramethylrhodamine (MTS-TAMRA; Biotium) for 7 min at 4 °C in a depolarizing solution (in mM: 120 K-Methanesulfonate (MES), 2 Ca(MES)$_2$, 10 HEPES; pH = 7.0). Alternate fluorophores attempted for

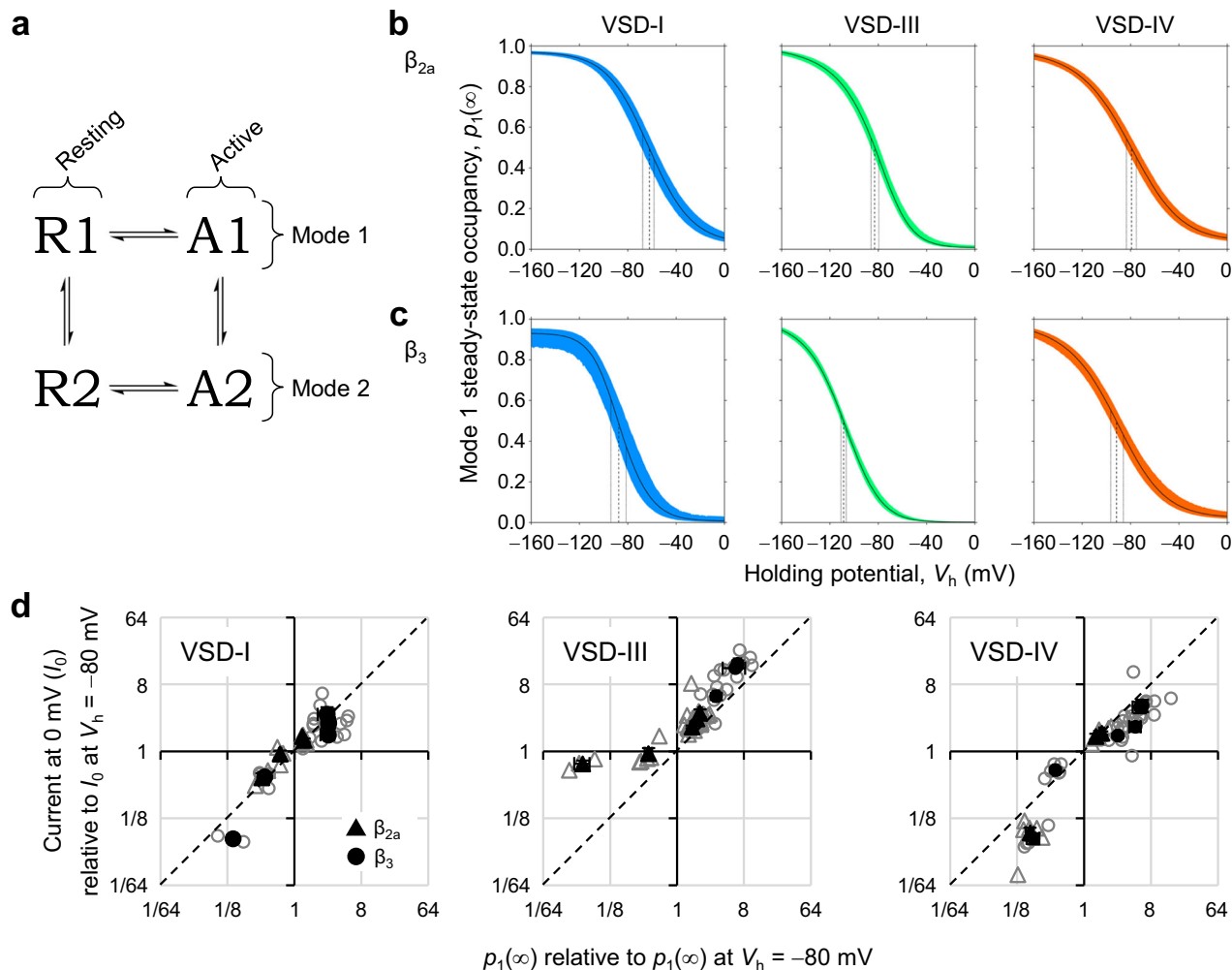

**Fig. 5 | VSD-I conversion is linked to VDI. a** The four-state model used to fit VCF data. Each VSD could achieve four conformations: R1: mode-1 resting state; A1: mode-1 active state; R2, A2: mode-2 resting and active states. Transitions are defined as: R→A: activation; A→R: deactivation; mode 1 → mode 2: conversion; mode 2 → mode 1: recovery. **b** The steady-state occupancy of mode-1 states ($p_1(\infty)$), i.e., $p_{R1}(\infty) + p_{A1}(\infty)$) plotted against the holding potential ($V_h$) in the presence of $\beta_{2a}$. Colored area: 95% confidence interval; vertical dashed lines point to the mean $V_{0.5}$; dotted lines: $V_{0.5}$ 95% confidence interval (parameters in Tables S2 and S3). **c** As in (**b**), now in the presence of VDI-favoring $\beta_3$-subunits: all conversions are facilitated,

occurring at more negative potentials (Table S2). **d** Mode-1 occupancy (relative to Mode 1 at $V_h = -80$ mV) plotted against relative current availability. The latter was calculated from inward current at 0 mV ($I_0$, as in the red traces in Fig. 3a) relative to $I_0$ at $V_h = -80$ mV. Open symbols are from individual cells, filled symbols are means. Mode-1 occupancy in VSD-I was statistically indistinguishable from current availability ($p = 0.967$, $n = 17$ cells), in contrast to VSD-III ($p = 0.0389$, $n = 26$) and VSD-IV ($p = 0.0102$, $n = 27$). Kolmogorov-Smirnov two-sided (two-sample) tests. Error bars are S.E.M.

VSD-II were: 10 μM tetramethylrhodamine-6-maleimide (TMR6M; AAT Bioquest) for 15 min at 4 °C, 20 μM tetramethylrhodamine-6-maleimide C6 (6-TAMRA C6 maleimide; AAT Bioquest) for 25 min at room temperature, 100 μM Alexa Fluor 488 $C_5$ maleimide (Alexa-488; Thermo Fisher Scientific) for 30 min on ice. Oocytes were rinsed in dye-free SOS following fluorescence labeling.

### Electrophysiological techniques

Oocytes were voltage-clamped under the cut-open oocyte Vaseline Gap (COVG) technique complemented with epifluorescence detection[24,27,29]. A CA-1B amplifier (Dagan Corporation) was used in COVG mode. Data were acquired at 25 kHz using a Digidata 1550B1 digitizer and pClamp 11.2.1 software (Molecular Devices). The optical set-up consisted of a BX51WI upright microscope (Olympus) with filters (Semrock BrightLine: exciter: FF01-531/40-25; dichroic: FF562-DiO2-25x36; emitter: FF01-593/40-25). The excitation light source was the M530L3 green LED (530 nm, 170 mW, Thorlabs) driven by a Cyclops LED driver (Open Ephys). For Alexa-488 experiments, the following filter set was used (Semrock BrightLine): exciter: FF01-482/

35-25; dichroic: FF506-DI03-25x36; emitter: FF01-524/24-25. The light source was a Thorlabs blue LED (490 nm, 205 mW, M490L4). A LUMPLNFL 40XW water immersion objective (Olympus; numerical aperture = 0.8, working distance = 3.3 mm) and SM05PD3A Si photodiode (Thorlabs) were used for fluorescence detection. Photocurrent was amplified with a DLPCA-200 current amplifier (FEMTO). Fluorescence emission and ionic currents were simultaneously recorded from the oocyte membrane isolated by the top chamber and low-pass-filtered at 5 kHz.

Prior to recordings, oocytes were injected with 100 nL of 100 mM BAPTA•4 K, 10 mM HEPES, pH=7.0 to prevent activation of endogenous $Ca^{2+}$- and $Ba^{2+}$-dependent $Cl^-$ channels. External solution (in mM): 120 NaMES, 2 Ba(MES)$_2$, 10 HEPES; pH=7.0. Internal solution (in mM): 120 K-glutamate, 10 HEPES; pH = 7.0. Intracellular micropipette solution: 3 M NaMES, 10 mM NaCl, 10 mM HEPES; pH=7.0. Oocytes were permeabilized using 0.1% saponin to gain low resistance intracellular access. Unless otherwise stated, oocytes were clamped at a holding potential of −80 mV. To evaluate the voltage dependence of channel activation, a series of 50 ms test pulses from −100 mV to 80 mV, in

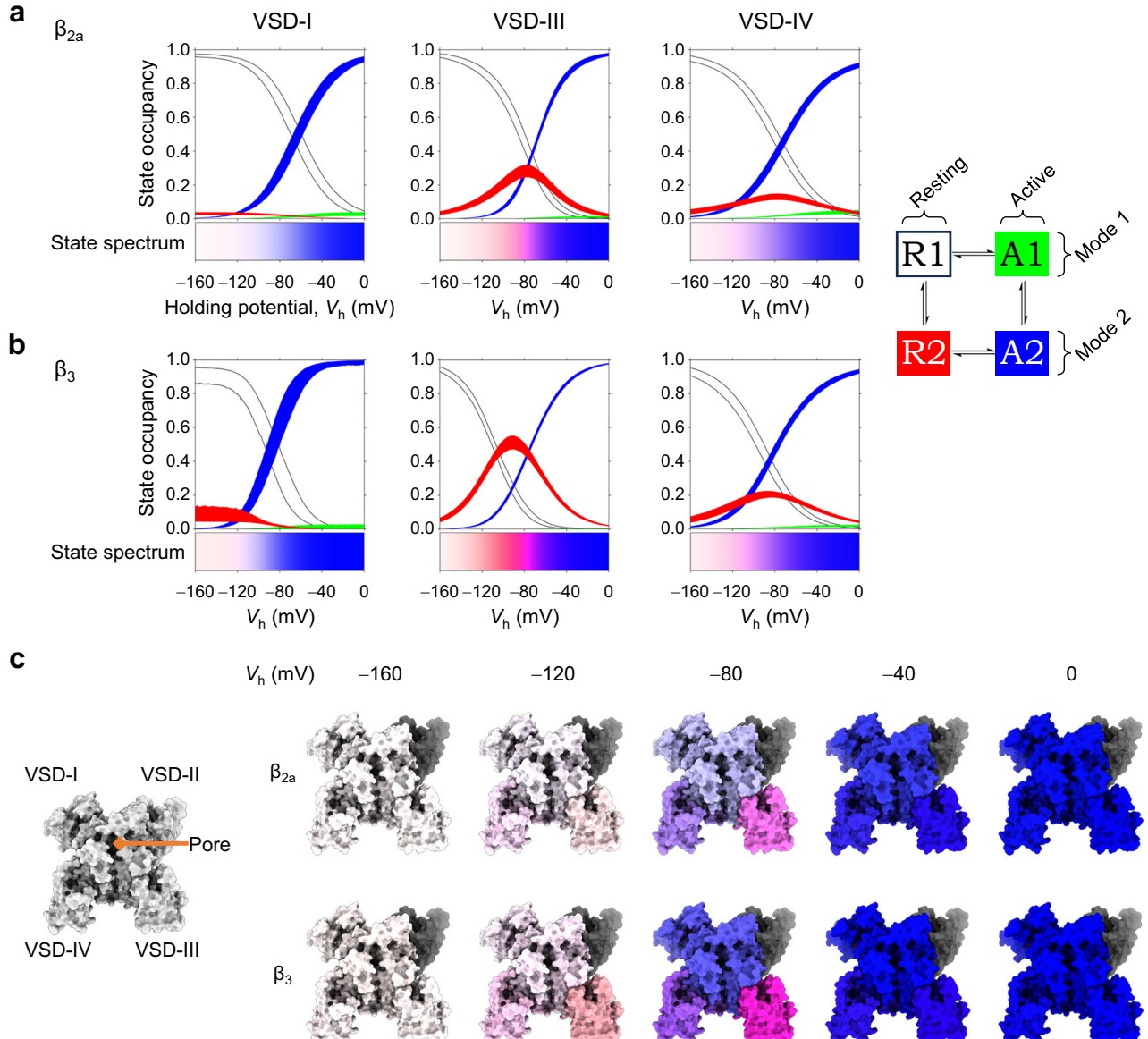

**Fig. 6 | The rich conformational palette of Ca$_V$2.1 at steady-state. a** Steady-state curves of all states (color code on the scheme on the right), plotted against the holding potential ($V_h$). Areas are 95% confidence intervals. VSD-I behaves as a canonical converting VSD, with states A1 and R2 being metastable (very low steady-state occupancy). The voltage-dependent R1-R2 conversions in VSD-III and VSD-IV result in stable R2 states around physiological resting potentials. The state spectra colorbars below encode the state occupancies into color information. **b** As in (**a**), for channels complexed with β$_3$. The VDI-favoring subunit alters the overall state spectra for all VSDs. **c** Color information in the state spectra was used to annotate the α$_{1A}$ surface (PDB: 8X90[38]) at different $V_h$ and β-subunits. The white-to-blue transitions illustrate the R1-to-A2 modal shift in VSD-I, while VSD-III and VSD-IV exhibit prominent red or purple hues, due to the stable occupancy of R2, colored red. The pore is colored white for closed, green for open and red/blue for inactivated; as inactivation best correlated with VSD-I modal shifts (Fig. 5d), it follows the same color. VSD-II is shown in gray. Supplementary Movie 1 is an animated version of this figure.

10 mV increments, was used. P/−6 subtraction was performed from −100 mV to reduce capacitive transients. To examine the voltage dependence of VSD-III and VSD-IV activation, 50 ms test pulses within the range of −200 mV to 60 mV, in 20 mV increments, were used. For VSD-I, an activating pulse of 100 ms was used unless otherwise stated, as fluorescence deflections did not achieve steady-state by 50 ms. 4 averages were performed to increase the signal-to-noise ratio of fluorescence signals. To evaluate different holding potentials ($V_h$), oocytes were clamped to each $V_h$ for 2 min to allow complete conversion of channels prior to running experimental protocols.

### Data analysis

The voltage dependence of channel opening was obtained from the peak tail current at $V_h = −80$ mV and fit to the single

Boltzmann function:

$$I_{tail}(V) = I_{tail,\,max}/\{1 + \exp[z\mathrm{F}(V_{0.5} − V)/(\mathrm{R}T)]\} \quad (1)$$

where $V$ was membrane potential, $I_{tail,max}$ was the maximal $I_{tail}$, $z$ was the valence, $V_{0.5}$ was the half-activation potential, F was the Faraday constant, R was the gas constant, and $T$ was temperature (294 K).

The voltage dependence of fluorescence deflections ($\Delta F$) was obtained from the averaged fluorescence signal during the last 5 ms of the test pulse. $\Delta F$ for VSDs III and IV were fit to a Boltzmann function:

$$\Delta F(V) = (\Delta F_{max} − \Delta F_{min})/\{1 + \exp[z\mathrm{F}(V_{0.5} − V)/(\mathrm{R}T)]\} + \Delta F_{min} \quad (2)$$

where $\Delta F_{max}$ and $\Delta F_{min}$ were the maximal and minimal $\Delta F$ asymptotes, respectively.

In the case of the $\Delta F(V)$ curve for VSD-I (Fig. 2f), and subsequent fittings of VSDs I, III, and IV at extended holding potentials (Figs. 3f, i & 4a,d,g,j), the sum of two Boltzmann distributions was used:

$$\Delta F(V) = \Delta F_{total} \cdot F_1 / \{1 + \exp[z_1 F(V_{0.5\_1} - V)/(RT)]\}$$
$$+ \Delta F_{total} \cdot (1 - F_1)/\{1 + \exp[z_2 F(V_{0.5\_2} - V)/(RT)]\} + \Delta F_{min\_1}$$
$$+ \Delta F_{min\_2}$$

(3)

where $\Delta F_{total}$ was the total fluorescence change ($\Delta F_{max\_1} + \Delta F_{max\_2} - \Delta F_{min\_1} - \Delta F_{min\_2}$) and $F_1$ was the fractional amplitude of the depolarized fluorescence component [($\Delta F_{max\_1} - \Delta F_{min\_1})/\Delta F_{total}$]. To help define the parameters, only cells with >1 $V_h$ were fit, and the following constraints were placed to reduce the number of free parameters:

1. Voltage-dependence parameters ($V_{0.5\_1}$, $z_1$, $V_{0.5\_2}$, and $z_2$) were constrained to be equal across fits of different $V_h$ for each cell.
2. $\Delta F_{min\_1}$ was constrained to be equal to $\Delta F_{max\_2}$ for each $V_h$, in each cell.

Fitting was performed by least squares using *Solver* in Microsoft Excel. Data are represented as mean ± S.E.M.

To determine the apparent voltage-dependence of VSD conversion from the Boltzmann fits (Figs. 3h, k & 4c, f, i, l), $F_1$ values from all cells and $V_h$ were pooled together and fit to the Boltzmann distribution:

$$F_1(V) = F_{1\_max}/\{1 + \exp[-zF(V_{0.5} - V_h)/(RT)]\}$$

(4)

$F_{1\_max}$ was fixed to 1 for VSD-I and VSD-III, and left as a free parameter for VSD-IV. Fitting was perfomed in Mathworks Matlab using *fit*. 95% confidence intervals were estimated using Mathworks Matlab *confint*.

To describe kinetic transitions among VSD states, we fit the fluorescence signals to a state-scheme mechanism, as previously[61,62]. Specifically here, a four-state model was constructed in MATLAB R2019a (MathWorks) representing transitions between a VSD active and resting states between modes 1 and 2. Activation transition rates (A1→R1 and A2→R2) were modelled as:

$$k = k_{eq} \cdot \exp[(V - V_{eq}) \cdot z \cdot \beta \cdot F/(RT)]$$

(5)

while deactivation transition rates (A1←R1 and A2←R2) as:

$$k = k_{eq} \cdot \exp[-(V - V_{eq}) \cdot z \cdot (1 - \beta) \cdot F/(RT)]$$

(6)

where $k_{eq}$, $V_{eq}$, $z$, and $\beta$ are shared free parameters. $k_{eq}$ is the rate at $V_{eq}$, $V_{eq}$ is the voltage where the forward and backward rates are equal, and $\beta$ is the portion of position of the energy barrier on the electric field. Conversion rate (R1→R2) was modelled as:

$$k = k_{con} \cdot \exp[V \cdot z \cdot \beta \cdot F/(RT)]$$

(7)

while recovery rates (R1←R2 and A1←A2) were modelled as:

$$k = k_{rec} \cdot \exp[-V \cdot z \cdot (1 - \beta) \cdot F/(RT)]$$

(8)

In this way, each conversion/recovery transition pair also shared four free parameters ($k_{con}$, $k_{rec}$, $z$, $\beta$), but this formulation was more easily adaptable to becoming voltage-independent by fixing $z$ to 0.

To obey microscopic reversibility, conversion rate (A1→A2) was calculated by:

$$k_{A1 \to A2} = k_{A1 \leftarrow A2} \cdot k_{R1 \leftarrow A1} \cdot k_{R2 \leftarrow R1} \cdot k_{A2 \leftarrow R2}/(k_{A2 \to R2} \cdot k_{R2 \to R1} \cdot k_{R1 \to A1})$$

(9)

This maneuver also reduced the number of free parameters by one, as $k_{con}$ did not have to be calculated for the A1↔A2 transitions.

Finally, to obey conservation of charge, the valence of the R2↔A2 transitions was also excluded as a free parameter, and was calculated by:

$$z_{R2 \leftrightarrow A2} = z_{R1 \leftrightarrow A1} + z_{A1 \leftrightarrow A2} - z_{R1 \leftrightarrow R2}$$

(10)

The model rates were formulated into a **Q**-matrix[63,64]. Briefly, **Q** was a square $4 \times 4$ matrix. Each element $q_{ij}$ contained the rate for the transition from state $i$ to state $j$. If there were no connection between states $i$ and $j$ then $q_{ij} = 0$. Each diagonal element was the negative sum of the off-diagonal elements in its row. In this way,

$$d\mathbf{p}(t)/dt = \mathbf{p}(t)\mathbf{Q}$$

(11)

where $\mathbf{p}(t)$ was a $1 \times 4$ vector of probability (occupancy) for each state. The MATLAB *ode15s* solver was used to calculate it. The voltage steps had a 43 µs time-constant to both emulate the COVG clamp speed and reduce stiffness. For initial conditions, background fluorescence calculations, and other calculations after fitting, the state occupancies at steady-state were calculated using:

$$\mathbf{p}(\infty) = \mathbf{u}^T(\mathbf{SS}^T)^{-1}$$

(12)

where $\mathbf{u}$ was a $4 \times 1$ unitary vector and $\mathbf{S}$ was $[\mathbf{Q}\ \mathbf{u}]$.

Finally, $4 \times 1$ vector $\mathbf{f}$ contained fluorescence levels of each state. State R1 fluorescence ($F_{R1}$) was fixed to 0. The fluorescence levels of states A1 and A2 ($F_{A1}$ and $F_{A2}$, respectively) were free parameters. State R2 fluorescence ($F_{R2}$) was constrained, thus, to impose conformational conservation:

$$F_{R2} = F_{A2} - F_{A1}$$

(13)

Fluorescence was then simulated as:

$$\Delta F = [\mathbf{p}(V, t) - \mathbf{p}(V_{h,n}, \infty)]\mathbf{f}b_n$$

(14)

where $V_{h,n}$ was the holding potential of the $n$th recording from the cell, and $b_n$ was a factor to account for fluorescence bleaching during the experiments, which reduced the $\Delta F$ amplitude. $b_1$ was fixed to 1, and $b_{n>1}$ had bounds 0 and 1.

Data from each cell with >2 $V_h$ were fit simultaneously. Rate optimization was performed by least squares, using the Bayesian adaptive direct search (BADS) machine-learning model-fitting algorithm[65].

The formulae for the R1-R2 rates did not contain a $V_{eq}$ parameter (Eqs. 6,7). When the R1-R2 equilibrium was voltage-dependent ($z > 0$, for VSD-III and VSD-IV), $V_{eq}$ was calculated after fitting, using:

$$V_{eq} = -\ln(k_{con}/k_{rec})RT/(Fz)$$

(15)

Finally, the equilibrium potential of modal shift was calculated iteratively using Matlab's *lsqcurvefit*, solving for the voltage where sum of the mode-1 steady-state occupancies was 0.5.

After several cells were fit, optimized, and calculated parameters were averaged: the geometric mean was used for rate-constant parameters ($k_{eq}$, $k_{con}$, $k_{rec}$), while the arithmetic mean was used for all

others. 95% confidence intervals were calculated by bootstrapping (Matlab *bootci*, 10000 iterations).

Mode-1 occupancy and available current correlations (Fig. 5d) were performed as follows: Available current was calculated using test-pulses to 0 mV with different $V_h$, which produced inward current according to channel availability (red traces in Fig. 3a). For each cell, the currents measured were normalized to the current with $V_h = -80$ mV, which was available in all recordings. Only cells whose $\Delta F$ were fit with the 4-state model were included in this dataset. Mode-1 occupancy was calculated as the sum of occupancies of states R1 and A1 using Eq. 12, and normalized to mode-1 occupancy with $V_h = -80$ mV. Two-sample Kolmogorov-Smirnov tests were used to compare the distributions of available channels and channels in mode 1.

State occupancies were converted into color information (state spectra, Fig. 6a, b) by first assigning the occupancies of states R2, A1, and A2 as red-green-blue (RGB, respectively) triplets. To encode the fourth state (R1) occupancy, the RGB triplets were converted into the hue-saturation-lightness (HSL) color model. The lightness values were then replaced by:

$$L = 0.5 + p_{R1}(\infty)/2 \tag{16}$$

Where $p_{R1}(\infty)$ is the steady-state occupancy of R1. In this way, when $p_{R1}(\infty) = 0$, the spectrum has medium lightness, allowing the underlying color to show; when $p_{R1}(\infty) = 1$, the spectrum has maximal lightness (white). The HSL triplets were then converted back into the RGB format, to construct the spectra or annotate the $Ca_V2.1$ structure (Fig. 6 & Supplementary Movie 1).

## Protein structure rendering

Structures of the human $Ca_V2.1$ $\alpha_{1A}$ subunit (PDB: 8X90[38]) were rendered on UCSF ChimeraX[66–68] and PyMOL (Schrödinger).

## Reporting summary

Further information on research design is available in the Nature Portfolio Reporting Summary linked to this article.

## Data availability

The data that support this study are available from the corresponding authors upon request. The source data underlying Figs. 2–5 are available at Zenodo. [https://doi.org/10.5281/zenodo.14615100]. The previously published structure of the human $Ca_V2.1$ $\alpha_{1A}$ subunit was used from PDB: 8X90[38]

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

## Acknowledgements

We thank members of the Pantazis group for useful discussions and members of the Elinder, Liin, and Pantazis groups for oocyte preparation. Funding: Lions Forskningsfond mot Folksjukdomar Ph.D. support (M.N.), NIH/NIGMS R35GM131896 (R.O.), start-up funds from the Linköping University Wallenberg Center for Molecular Medicine / the Knut and Alice Wallenberg Foundation (A.P.), Hjärnfonden (The Swedish Brain Foundation) grants and FO2022-0219 (F.E.), FO2022-0003 and FO2023-0025 (A.P.), Hjärt-Lung Fonden (The Swedish Heart-Lung Foundation) 20210596 (F.E.), Vetenskapsrådet (The Swedish Research Council) grants 2020-01019 (F.E.), and 2019-00988 and 2022-00574 (A.P.).

## Author contributions

K.W., M.N., and M.A. performed experiments. K.W., M.N., and A.P. performed analysis. R.O. and F.E. contributed reagents and methods. K.W., M.N., F.E., and A.P. wrote the manuscript. All authors contributed to manuscript review and editing.

## Funding

## Competing interests

The authors declare no competing interests.
