## [Transparent Peer Review file · Nature Communications]

A Rich Conformational Palette Underlies Human Ca_v2.1-Channel Availability

Corresponding Author: Dr Antonios Pantazis

Version 0:

Reviewer comments:

Reviewer #1

(Remarks to the Author)

Summary

In this manuscript, Wang and collaborators present interesting results assessing Ca_v2.1 Ca²⁺ channels voltage sensor domains (VSDs) conformational changes using voltage clamp fluorometry performed in the *Xenopus* oocytes expression system. In addition to the characterization of individual VSDs' voltage dependence of activation, the authors assessed the role of each VSD for voltage-dependent inactivation and the impact of two different Ca_vβ subunits, important fine-tuners of the channel's activity, on VSD conformational change. Senior authors are experts in the field and pioneers of implementing voltage clamp fluorometry in voltage-gated potassium and calcium channels. The manuscript concludes with a simple but powerful mathematical model that catches the essence of the Ca_v2.1 gating mechanism and allows them to test their predictions. They also provide additional structural modeling that would be valuable for testing new therapeutic development and missense mutation pathogenicity prediction. Here, they present a well-written story that is easy to read and significant for voltage sensing mechanisms in channels operated by voltage. The experiments are well performed with adequate replicates, and the data is adequately analyzed and suitably interpreted. However, some aspects of the manuscript must be handled before recommending publication.

The authors may wish to take into consideration the following points:

Major comments

Introduction:

"A prolonged depolarization or train of action potentials cause Ca_v2.1 voltage-dependent inactivation (VDI)". Repetitive stimulation can also lead to voltage-dependent facilitation of HVA channels, and P/Q channels are known to have this form of modulation (see Currie KP, Fox AP. Differential facilitation of N- and P/Q-type calcium channels during trains of action potential-like waveforms. *J Physiol.* 2002 Mar 1;539(Pt 2):419-31. doi: 10.1113/jphysiol.2001.013206. PMID: 11882675; PMCID: PMC2290166), which depends on direct channel regulation mediated by neurotransmitters' action on GPCR and activating G-proteins. Please include this in the introduction.

Results:

In this work, the time course and voltage dependence of VSD individual signals (F-V) are compared with that of ionic current using barium as charge carrier (IBa) but not with the trajectory of gating currents and voltage dependence of the integrated charge (Q-V). Yet, in previous manuscripts from the same group, fluorometric signals are compared to gating current (IQ).

Were any measurements of gating currents and charge movement (Q-V) analyses performed?

What would IQ's time course look like compared to individual VSD molecular rearrangement?

Would neutralizing charged residues in VSD2, a voltage sensor regarded as immobile in this report, impact total IQ?

Such a comparison would be helpful, especially for VSD-I, which seems so slow to activate, vs. VSD-III or VSD-IV.

Figure 2: VSD-I, -II, and -III conformational rearrangement from -200 mV to -80 mV represent almost 50% of overall fluorescent change. At this voltage, IBa would not occur, but is any IQ manifestation observable (inward or outward direction)? If not, how can the authors reconcile the fact that these VSDs, which possess the "charge particles" of the channel, can move within the lipid bilayer (as seen through fluorescent signal) without producing substantial IQ?

Regarding VSD-II immobility. The authors explored several strategies, including (a) probing most of the S3-S4 linker, (b) trying different fluorophores, (c) removing a tryptophan that might quench the nearby fluorophore, (d) using a different

complement of auxiliary subunits, (e) neutralizing counter-charges that could stabilize the S4II resting state and (f) perturbing a PIP2 binding site.

However, one additional aspect that would be worth exploring is the role of the lipidic environment. Defining the function of proteins requires that their activity be measured under controlled conditions *in vitro* that replicate the native environment (*in vitro veritas*). Ideally, membrane proteins, particularly ion channels, must be studied in their native environment to be fully functional. It is known that the composition of membranes varies between species, cell types, and organelles, and even the distribution of lipids is asymmetric between the membrane outer and inner leaflets. The lipid environment is not only optimal for protein-membrane insertion but also for fine-tuning its function. Indeed, some estimates of the relative abundance of lipids in *Xenopus* suggest that glycerophospholipids (PC and PI) represent 43 % and sphingomyelin (SM) 26 % of total lipids (Hill WG, Southern NM, MacIver B, Potter E, Apodaca G, Smith CP, Zeidel ML. Isolation and characterization of the *Xenopus* oocyte plasma membrane: a new method for studying the activity of water and solute transporters. *Am J Physiol Renal Physiol.* 2005 Jul;289(1):F217-24. doi: 10.1152/ajprenal.00022.2005. Epub 2005 Mar 1. PMID: 15741609), whereas in the human brain PC and PI constitute ~18% and SM ~7% of total lipids (see: Osetrova M, Tkachev A, Mair W, Guijarro Larráz P, Efimova O, Kurochkin I, Stekolshchikova E, Anikanov N, Foo JC, Cazenave-Gassiot A, Mitina A, Ogurtsova P, Guo S, Potashnikova DM, Gulin AA, Vasin AA, Sarycheva A, Vladimirov G, Fedorova M, Kostyukevich Y, Nikolaev E, Wenk MR, Khrameeva EE, Khaitovich P. Lipidome atlas of the adult human brain. *Nat Commun.* 2024 May 25;15(1):4455. doi: 10.1038/s41467-024-48734-y. PMID: 38796479; PMCID: PMC11127996).

The authors may want to explore this possibility by modifying the lipidic environment of the oocyte. We understand that this is not an easy task. Still, at least some other groups have considered these issues and methods (See: Ivorra I, Alberola-Die A, Cobo R, González-Ros JM, Morales A. *Xenopus* Oocytes as a Powerful Cellular Model to Study Foreign Fully-Processed Membrane Proteins. *Membranes (Basel).* 2022 Oct 11;12(10):986. doi: 10.3390/membranes12100986. PMID: 36295745; PMCID: PMC9610954. At the very least, this should be addressed in the discussion.

Using different fluorescent reporters to track the same cysteine could induce different fluorescent profiles (for example see: Wojciechowski MN, McKenzie CE, Hung A, Kuanyshbek A, Soh MS, Reid CA, Forster IC. Different fluorescent labels report distinct components of spHCN channel voltage sensor movement. *J Gen Physiol.* 2024 Aug 5;156(8):e202413559. doi: 10.1085/jgp.202413559. Epub 2024 Jul 5. PMID: 38968404; PMCID: PMC11223168). Here, the authors compared different dyes for VSD-II to see if other reporters would detect any motion from this segment. This important control supports the author's conclusion of VSD-II "non-motion." Yet, this raises a new concern: did the authors compare different dyes for VSD-I, -III, and -IV while labeling the same cysteine and using a similar depolarization protocol? For example, would VSD-I change of fluorescence be faster with other dyes? Also, would using another dye change the individual VSD voltage dependence?

Figure S1: Why only show the optical signal from -80 mV to +80 mV depolarization rather than from more hyperpolarized voltage (i.e., -200 mV to +80 mV)? A larger ΔV_m excursion could resolve subtle VSD-II fluorescence changes or allow observation of potential "F2" components.

Figure S2: "Introduction of point mutations in VSD-II (and fluorescent labeling with MTS-TAMRA) did not substantially alter the voltage-dependence of pore opening compared to wild-type channels". Some constructs may show a small shift in voltage dependence, implying a role of VSD-II in channel gating (see, for example, CaV2.1 WT $V_{0.5} = 2.2\text{mV}$ while I579C $V_{0.5} = -8.9\text{mV}$). Was any statistical analysis performed on $V_{0.5}$ of WT vs. VSD-II engineered channels? It would be valuable to include it in the table.

Figure 3b: the authors used a protocol in which the cell is depolarized for 2 min to V_h and then brought to $V_m = -80\text{ mV}$ for 100 ms and then to the test pulse V_m . Was any change of VSD fluorescence observed between V_h to $V_m = -80\text{mV}$? If so, was a 100ms interval sufficient to fully reach steady state fluorescence before depolarizing the cell to V_m test pulse?

Methods: "Four averages were performed to increase the signal-to-noise ratio of fluorescence signals." Do these averages have been performed for each voltage command? If 20 depolarizations are applied on the same cell, this will imply a total of 80 traces have been measured. Is there any MTS-TAMRA bleaching during this process? If so, how was it accounted for?

Minor comments

Results: "To optically track the movements of individual VSDs under physiologically relevant conditions, we used VCF". The channel is reconstituted with CACNA2D1 and CACNB-2A or -B3 in a *Xenopus* oocyte. The authors mention later that other accessory subunits and/or PIP2 could regulate the channel. Thus, the statement "physiologically relevant conditions" seems overstated since only the essential components of the channel are present in the oocyte.

Figure 2: It would be important for the readers if the authors explain why the direction of fluorescence change is opposite for VSD-IV compared to VSD-I or VSD-III while the same dye is used for all VSD tracking.

Figure 2f to 2h: consider presenting the fitting parameters in a table for easier comparison of F1 and F2 activation for each VSD.

Figure 3b: Consider including the V_h before returning to -80 mV and then to the test pulse in the depolarization protocol cartoon. That would make the protocol easier to understand without looking at the methods section.

Methods: "P/6 subtraction was performed to reduce capacitive transients". The rearrangement of some VSDs occurred at a relatively low voltage (below -80 mV). At which holding voltage P/6 was run?

Reviewer #2

(Remarks to the Author)

In this study, Kaiqian Wang and collaborators under the lead of Antonios Pantazis studied the structural dynamics of the four voltage-sensing domains (VSD) of CaV2.1 calcium channels and derived information about their specific roles in channel gating and voltage-dependent inactivation (VDI). This study addresses a timely and important scientific problem, relevant for both channel biophysics and basic neurosciences.

To tackle this problem, the authors apply voltage-clamp fluorometry of genetically altered and labeled CaV2.1 channels expressed in oocytes, a highly sophisticated technique that allows to track the voltage-dependence and kinetics of the conformational rearrangements of individual VSDs in response to depolarization. The experiments are of the highest standards, the data are convincing and clearly presented in the figures, tables and supplementary information.

Their recordings reveal distinct voltage-dependent activation of VSDs I, III and IV. Consistent with some structural data finding VSD II locked in the resting state, VSD II did not move in response to depolarization. This finding is backed up by numerous controls excluding the possibility that the observed lack of VSD II movement resulted from a technical problem or was dependent on subunit or lipid interactions.

Most intriguingly, in their voltage-clamp fluorometry experiments VSD I displayed a bi-modal voltage dependence, one of which (F1) closely corresponded to channel opening. Different holding potentials favored one or the other state, indicating that this mode shift of VSD I underlies VDI. Co-expressing beta 3 instead of beta 2a shifted the voltage-dependence of the two modes. The data were fitted well with a four-state model. Analyzing the state transitions allowed the authors further characterization of the molecular mechanisms underlying VDI. In conclusion, the results of the study convincingly demonstrate that VSD I controls channel opening and its conversion into mode 2 results in VDI of CaV2.1.

Furthermore, the data analysis with the four-state model reveal numerous intriguing aspects related to CaV2.1 both in experimental situations and in real life. Some of these deserve further (future) investigations, other's could be discussed here. For example, relating to Figure 6, in what state are the four VSDs at the resting potential of a typical neuron, how do these sequentially transition on depolarization / channel opening? Or, which states do the four VSDs most likely occupy in preparations used for cryo-EM structure analysis?

Overall, this is a fine study, ready to be published. A couple of questions could be addressed for clarity purposes:

1. The reason why the authors use the heterologous beta 2a is clear. But why did the authors use beta 3 for comparison, rather than beta 4, which represents the foremost partner of CaV1.2 in CNS neurons?
2. VSD-I interconversion from F1 (available) to F2 (inactivated) is a highly plausible explanation of the bi-modal voltage-dependence of activation. Classical charge interconversion of LTCC is characterized by the constancy of total gating charges moved in the two modes. Can the authors analyze their fluorometry data to demonstrate that the total amount of conformational conversion remains constant at the different holding potentials? In the normalized plots in Fig. 3 this information is lost.

Minor:

Line 74 For an optical recording technique, the expression, "faint ΔF signal" can be misunderstood to mean that the fluorescence signal itself was faint (rather than the delta), thus implying a technical problem.

Lines 682-684 The cited references refer to CaV2.2 and 2.3 channels, not to CaV2.1, although the mechanism may be the same. Make clear!

Suppl. Data:

Line 58 "normalized (a)" should be (c)

Reviewer #3

(Remarks to the Author)

Version 1:

Reviewer comments:

Reviewer #2

(Remarks to the Author)

I had little to criticise to begin with. In the revised version the authors addressed all my comments to my full satisfaction. This is a fine study providing lots of interesting insights in the molecular mechanisms of voltage sensing and channel gating. I have no further concerns.

Reviewer #3

(Remarks to the Author)

REVIEWER COMMENTS

Reviewer #1 (Remarks to the Author):

Thank you for your insightful comments and useful feedback. We have addressed all your concerns below (in blue italics), and we hope our responses are to your satisfaction.

Summary

In this manuscript, Wang and collaborators present interesting results assessing CaV2.1 Ca²⁺ channels voltage sensor domains (VSDs) conformational changes using voltage clamp fluorometry performed in the *Xenopus* oocytes expression system. In addition to the characterization of individual VSDs' voltage dependence of activation, the authors assessed the role of each VSD for voltage-dependent inactivation and the impact of two different CaV β subunits, important fine-tuners of the channel's activity, on VSD conformational change. Senior authors are experts in the field and pioneers of implementing voltage clamp fluorometry in voltage-gated potassium and calcium channels. The manuscript concludes with a simple but powerful mathematical model that catches the essence of the CaV2.1 gating mechanism and allows them to test their predictions. They also provide additional structural modeling that would be valuable for testing new therapeutic development and missense mutation pathogenicity prediction. Here, they present a well-written story that is easy to read and significant for voltage sensing mechanisms in channels operated by voltage. The experiments are well performed with adequate replicates, and the data is adequately analyzed and suitably interpreted.

Thank you for your positive appraisal of our work.

However, some aspects of the manuscript must be handled before recommending publication. The authors may wish to take into consideration the following points:

Major comments

Introduction:

"A prolonged depolarization or train of action potentials cause CaV2.1 voltage-dependent inactivation (VDI)". Repetitive stimulation can also lead to voltage-dependent facilitation of HVA channels, and P/Q channels are known to have this form of modulation (see Currie KP, Fox AP. Differential facilitation of N- and P/Q-type calcium channels during trains of action potential-like waveforms. *J Physiol.* 2002 Mar 1;539(Pt 2):419-31. doi: 10.1113/jphysiol.2001.013206. PMID: 11882675; PMCID: PMC2290166), which depends on direct channel regulation mediated by neurotransmitters' action on GPCR and activating G-proteins. Please include this in the introduction.

Done. p.2 lines 24-25.

Results:

In this work, the time course and voltage dependence of VSD individual signals (F-V) are compared with that of ionic current using barium as charge carrier (IBa) but not with the trajectory of gating currents and voltage dependence of the integrated charge (Q-V). Yet, in previous manuscripts from the same

group, fluorometric signals are compared to gating current (IQ).

Were any measurements of gating currents and charge movement (Q-V) analyses performed?

We could not perform gating current (IQ) measurements in this work. P/Q-type IQ currents are notoriously difficult to detect, by ourselves and also others in the Ca_v field. For this reason, we dedicated several years implementing VCF to $Ca_v2.1$, first in the murine and then in the human isoform.

We are only aware of one publication with IQ measurements in $Ca_v2.1$: Barrett et al. (2005) J Biol Chem <https://doi.org/10.1074/jbc.M502223200> They only occur at depolarized voltages. In the methods, the authors mention: “residual linear capacitive and leak currents were subtracted by the -P/4 method”. Our VCF investigation revealed that much $Ca_v2.1$ VSD movements occur at negative voltages, near the resting membrane potential. So likely the subtraction pulses (and even standard capacitance compensation) would remove a large portion of the charge movement occurring at negative voltages.

Subtraction pulses in positive voltages would also be problematic. We showed that $Ca_v2.1$ VSDs undergo relatively rapid conversion which radically alter their biophysical properties. Performing subtraction pulses at positive voltages would produce conversion and confound the estimation of IQ voltage-dependence. And even in this case, the capacitive transient compensation would interfere with charge movement measurements.

Last but not least, another reason to exclude IQ measurements is our unpublished work on yet another member of the Ca_v family: we show that pore blockade by cations can substantially alter VSD voltage dependence (as resolved by VCF). We have presented these data at previous Biophysical Society meetings, but we cannot reveal more here, because this rebuttal will be published. In our opinion, it is reason enough to prefer VCF, which discloses VSD movement in conducting channels.

To acknowledge IQ measurements in our manuscript, we have added this text in the beginning of the discussion (p.5, lines 6-10):

“VSDs undergo activation transitions over a very broad range of voltages (fig.2i). A study using gating-current measurements reported charge movement (i.e., VSD transitions) at only depolarized voltages³⁸; we believe a reason for this discrepancy is that the capacitance compensation and “P/N” subtraction protocols used to measure gating currents likely hindered the detection of charge movement at negative voltages.”

What would IQ's time course look like compared to individual VSD molecular rearrangement?

Would neutralizing charged residues in VSD2, a voltage sensor regarded as immobile in this report, impact total IQ?

Such a comparison would be helpful, especially for VSD-I, which seems so slow to activate, vs. VSD-III or VSD-IV.

We agree, this alternative method of detecting charge movement might have offered some additional information. As mentioned above, we could not detect IQ currents from $Ca_v2.1$, and this method has considerable caveats due to the necessary subtraction protocols (which may either subtract gating charge or induce conversion) and pore blockade.

Figure 2: VSD-I, -II, and -III conformational rearrangement from -200 mV to -80 mV represent almost 50% of overall fluorescent change. At this voltage, I_{Ba} would not occur, but is any IQ manifestation

observable (inward or outward direction)? If not, how can the authors reconcile the fact that these VSDs, which possess the “charge particles” of the channel, can move within the lipid bilayer (as seen through fluorescent signal) without producing substantial IQ?

Indeed, it is very difficult to observe IQ in $Ca_v2.1$ channels relative to other Ca_v channels, where gating currents are readily observable together with ionic currents. We believe that (at least part of) the reason is that there are substantial VSD movement at negative voltages (as shown by the VCF experiments), so IQ is subtracted by both the capacitance compensation circuit and the P/-6 protocol in our records.

This is why VCF signals are a better measurement of VSD activation since they are not affected by capacitance compensation and subtraction maneuvers. We are confident that the VCF signals faithfully reflect specific VSD movements because (1) they are not observed in channels without introduced Cysteines (Fig.2e) and (2) they exhibit properties consistent with VSD conformational changes: sigmoidal voltage-dependence, exponential kinetics, conversion dependent on the holding potential, and regulation by beta subunits.

Regarding VSD-II immobility. The authors explored several strategies, including (a) probing most of the S3-S4 linker, (b) trying different fluorophores, (c) removing a tryptophan that might quench the nearby fluorophore, (d) using a different complement of auxiliary subunits, (e) neutralizing counter-charges that could stabilize the S4II resting state and (f) perturbing a PIP2 binding site.

However, one additional aspect that would be worth exploring is the role of the lipidic environment. Defining the function of proteins requires that their activity be measured under controlled conditions in vitro that replicate the native environment (in vitro veritas). Ideally, membrane proteins, particularly ion channels, must be studied in their native environment to be fully functional. It is known that the composition of membranes varies between species, cell types, and organelles, and even the distribution of lipids is asymmetric between the membrane outer and inner leaflets. The lipid environment is not only optimal for protein-membrane insertion but also for fine-tuning its function. Indeed, some estimates of the relative abundance of lipids in *Xenopus* suggest that glycerophospholipids (PC and PI) represent 43 % and sphingomyelin (SM) 26 % of total lipids (Hill WG, Southern NM, MacIver B, Potter E, Apodaca G, Smith CP, Zeidel ML. Isolation and characterization of the *Xenopus* oocyte plasma membrane: a new method for studying the activity of water and solute transporters. *Am J Physiol Renal Physiol.* 2005 Jul;289(1):F217-24. doi: 10.1152/ajprenal.00022.2005. Epub 2005 Mar 1. PMID: 15741609), whereas in the human brain PC and PI constitute ~18% and SM ~7% of total lipids (see: Osetrova M, Tkachev A, Mair W, Guijarro Larraz P, Efimova O, Kurochkin I, Stekolshchikova E, Anikanov N, Foo JC, Cazenave-Gassiot A, Mitina A, Ogurtsova P, Guo S, Potashnikova DM, Gulin AA, Vasin AA, Sarycheva A, Vladimirov G, Fedorova M, Kostyukevich Y, Nikolaev E, Wenk MR, Khrameeva EE, Khaitovich P. Lipidome atlas of the adult human brain. *Nat Commun.* 2024 May 25;15(1):4455. doi: 10.1038/s41467-024-48734-y. PMID: 38796479; PMCID: PMC11127996).

The authors may want to explore this possibility by modifying the lipidic environment of the oocyte. We understand that this is not an easy task. Still, at least some other groups have considered these issues and methods (See: Ivorra I, Alberola-Die A, Cobo R, González-Ros JM, Morales A. *Xenopus* Oocytes as a Powerful Cellular Model to Study Foreign Fully-Processed Membrane Proteins. *Membranes* (Basel). 2022 Oct 11;12(10):986. doi: 10.3390/membranes12100986. PMID: 36295745; PMCID: PMC9610954. At the very least, this should be addressed in the discussion.

*Thank you for your extended consideration on the effect of the lipidic composition of the plasma membrane. It is not something that we readily consider because the *Xenopus* oocyte expression*

system has been used for decades for the study of membrane proteins. This includes ion channels, receptors and transporters from bacteria, yeast, plants; as well as intracellular ion-channels. We appreciate this consideration and we have written about membrane composition in our manuscript (p.6, lines 16-19):

“Whether VSD-II can activate under different conditions (such as the signaling milieu or additional protein partners) is an outstanding question. Another consideration is that the lipidic composition of the plasma membrane in our heterologous expression system (Xenopus laevis oocytes) differs from that in human neurons^{48,49}.”

We also agree in principle with the “in vitro veritas” philosophy and this is why we favor the study of conducting-channel conformational dynamics in cellula.

Here are a few more of our considerations, for the sake of discussion:

(1) Oocytes have substantial benefits for VCF, affording the ability to record strong signals with high bandwidth and low background; (2) oocytes are an ideal expression system for the study of Ca_v channels, given their low run-down and high stability. These are major advantages especially for this study on long-term VSD conformational dynamics, which included hour-long experiments on the same cell. Combined with (3) using the fast cut-open oocyte Vaseline gap voltage-clamp, we firmly believe that this is the best system to experimentally study VSD conformational dynamics available to-date.

Could the oocyte lipidic composition affect VSD-II or other movements? We don't think so, because the voltage-dependence of $Ca_v2.1$ conductance is similar, in our system, to that of other expression systems—and as reported for other ion channels. Second, VSD-II was found to be “locked down” in a resting conformation in published structures which did not use channels expressed by Xenopus oocytes.

We would like to conclude by stating that we are still perplexed by the lack of VSD-II movements, in both $Ca_v2.1$ and $Ca_v2.2$ (Nilsson et al., Sci Adv 2024). In follow-up work, we will study pathogenic variants in VSD-II with VCF, and we are open to collaborating with groups also interested in solving this conundrum.

Using different fluorescent reporters to track the same cysteine could induce different fluorescent profiles (for example see: Wojciechowski MN, McKenzie CE, Hung A, Kuanyshbek A, Soh MS, Reid CA, Forster IC. Different fluorescent labels report distinct components of spHCN channel voltage sensor movement. J Gen Physiol. 2024 Aug 5;156(8):e202413559. doi: 10.1085/jgp.202413559. Epub 2024 Jul 5. PMID: 38968404; PMCID: PMC11223168). Here, the authors compared different dyes for VSD-II to see if other reporters would detect any motion from this segment. This important control supports the author's conclusion of VSD-II “non-motion.” Yet, this raises a new concern: did the authors compare different dyes for VSD-I, -III, and -IV while labeling the same cysteine and using a similar depolarization protocol? For example, would VSD-I change of fluorescence be faster with other dyes? Also, would using another dye change the individual VSD voltage dependence?

Indeed, we have previous experience that different ΔF can sometimes be observed with different fluorophores (Savalli et al., PNAS 2006; PMID: 16895996). As in any VCF investigation of a new channel, we have evaluated different labeling positions and fluorophores. We try to determine the best compromise between consistent fluorescence signals and minimal gating effects.

We have performed extensive tests with the alternative TMR-6'-M fluorophore. A benefit of TMR-6'-M was that it reported no ΔF in WT channels (without introduced Cys), so we tried it extensively. Using this fluorophore resulted in very faint VSD-I signals:

For VSD-I, we tried TMR-6'-M at positions E188C and T187C. It showed extremely faint signals that precluded further analysis:

[representative traces from a1A(E188C) / b2a / a2d labeled with TMR-6'-M]

This is somewhat expected from previous work on Ca_v channels: MTS-TAMRA generally showed stronger and more consistent ΔF , in our published work on $Ca_v1.2$ (Pantazis et al., 2014 PNAS & Savalli et al., 2016 J Gen Physiol), $Ca_v1.1$ (Savalli et al., 2021 J Gen Physiol) and $Ca_v2.2$ (Nilsson et al., 2024 Sci Adv), as well as more, yet-unpublished work on $Ca_v1.1$ and other Ca_v s. We think this is due to a combination of (1) the extensive extracellular pore loops that may reduce the accessibility of S4 cysteines; and (2) the reduced reactivity of maleimide dyes compared to MTS for cysteine.

However, in the case of VSD-III and VSD-IV, TMR-6'-M provided more useable DF , and its properties were practically identical to those of MTS-TAMRA. We opted to continue our investigation (over different V_h and beta subunits) with MTS-TAMRA, for consistency.

In addition, we have scanned several Cysteine positions at the start of the study with MTS-TAMRA:

VSD-I	E188C , F189C, L191C, T187C, D190C
VSD-III	I1339C, N1340C , T1341C, I1342C
VSD-IV	N1652C , N1653C, F1654C, I1655C, N1656C, S1658C

The positions in bold were the best compromise for good expression level, minimal gating perturbation, good and consistent DF signal, and we opted to continue with these for in-depth investigation (different V_h and beta subunits).

In summary, we have considered and extensively investigated alternative positions and fluorophores. The work we show here includes the most consistent implementation of VCF on $Ca_v2.1$ that we could achieve, to the best of our experience and efforts.

Two final considerations:

- 1) While individual VSD properties may be somewhat affected by the labeling, we believe these perturbations are small, as they come from conducting channels with wild-type-like voltage-dependence (see curves in Fig.2)

2) Experiments studying the effects of V_h , beta subunits and VDI (i.e. channel availability, Fig.5d) are all from channels with the same Cys and fluorophore, so they are internally controlled.

Figure S1: Why only show the optical signal from -80 mV to +80 mV depolarization rather than from more hyperpolarized voltage (i.e., -200 mV to +80 mV)? A larger ΔV_m excursion could resolve subtle VSD-II fluorescence changes or allow observation of potential “F2” components.

First, just to point out that, in principle, any VSD-II “F2” movement occurring over negative voltages would be very unlikely to be connected to VSD opening, so our conclusions on the lack of VSD-II relevance to pore opening would not be affected by such a result.

Second, if VSD-II activation (F1 or F2) did occur at negative potentials, then VSD-II would be expected to be in an active conformation at 0 mV, contradicting the structural biology results.

Traces from the -80 to +80 mV pulses are shown in fig.S1 because this was the pulse most consistently used. We agree that if a fluorescence change were there, it could be more obvious with a bigger voltage-step. However, we can detect subtle ΔF from VSD-II positions (fig.2 and some traces in fig.S1). The problem is not lack of detection; it's that the faint signal is qualitatively similar to that from WT channels, i.e. linear with respect to voltage. So our view is that these signals, while detectable, are artefactual.

We also agree, that pulsing more extensively might resolve VSD-II movements with extremely shifted voltage dependencies. Relevant to your concern, one question we had in mind was “is VSD-II permanently converted?”

First, we tried holding at -120 and even -160 mV, to recover more channels from inactivation (and enhance any putative “F1” component), but the outcome was the same—no significant ΔF resolved from VSD-II. A representative example showing current and fluorescence traces from the same cell labelled in VSD-II at different holding voltages is shown below:

[Note that holding at -120 mV results in bigger currents (recovery from inactivation), but no change in ΔF . In this cell, holding at -160 mV resulted in large leak currents, but we are confident that ΔF would have been detectable because the oocyte was otherwise intact]

Likewise, we could not detect significant ΔF when we held cells to $+40$ mV. In these experiments, we hoped that VSD-II movements would be detectable if we pushed VSD-II to convert, to accentuate any “F2” component.

In short, we committed a substantial amount of time doing all we could to detect signals from VSD-II. We began this investigation in 2019, before the first Ca_v2 structures were published in 2021 (which showed a “locked down” VSD-II), so we were very intent on ensuring that the lack of ΔF was more than a negative result.

Finally, as an additional point supporting the work on $Ca_v2.1$, our extended voltage steps in $Ca_v2.2$ VSD-II also showed no ΔF (Nilsson et al. 2024 Sci Adv, Figs.S1 & S2).

Figure S2: “Introduction of point mutations in VSD-II (and fluorescent labeling with MTS-TAMRA) did not substantially alter the voltage-dependence of pore opening compared to wild-type channels”. Some constructs may show a small shift in voltage dependence, implying a role of VSD-II in channel gating (see, for example, $Ca_v2.1$ WT $V_{0.5} = 2.2$ mV while I579C $V_{0.5} = -8.9$ mV). Was any statistical analysis performed on $V_{0.5}$ of WT vs. VSD-II engineered channels? It would be valuable to include it in the table.

Thank you for carefully considering our results. We did not perform a statistical analysis, because we did not design our experiments to allow for a valid analysis. Specifically, we did not perform WT controls on every oocyte batch where a mutant was tested, so statistical power is limited. We have amended our results text, as it unintentionally implied a statistical investigation (p.3, lines 18-19):

“Indeed, none of the mutations used in the above studies significantly altered the numerous VSD-II mutations tested did not result in a consistent or substantial alteration of the voltage-dependence of pore opening (fig.S2)”

A few more considerations:

Our experiments were designed to facilitate the testing of numerous conditions with VCF in a timely manner (it still took us several years to perform this work). Of course, we did perform WT labeled VCF controls for any mutant condition where a hint of ΔF was observed, as this was the focus of our investigation.

While our main question was on VSD-II movements (which we addressed with VCF), we included fig. S2 to show that there is no consistent effect on gating across a large number of mutants tested. Our interpretation of this lack of coherency is this: since VSD-II is resolved to be in close contact with pore and gate regions of the channel (as expected), subtle changes in structure or folding produced by mutations may affect gating in different ways, without need for VSD-II voltage-dependent movements. This interpretation is consistent with VCF and cryo-EM results across the Ca_v2 -channel family.

Figure 3b: the authors used a protocol in which the cell is depolarized for 2 min to V_h and then brought to $V_m = -80$ mV for 100 ms and then to the test pulse V_m . Was any change of VSD fluorescence observed between V_h to $V_m = -80$ mV? If so, was a 100ms interval sufficient to fully reach steady state fluorescence before depolarizing the cell to V_m test pulse?

Thank you for your question, which prompted us to clarify fig.3. First, 100 ms is more than sufficient to observe transitions between R1 and A1 states (seen when pulsing with $V_h = -80\text{mV}$ in figs 2a & 3b,d).

Fig. 3c also shows a pulse from -80 mV to $+40\text{ mV}$; but here, the cell was held at -80 for just 100 ms (V_h was $+40$). There was no fluorescence change from the pulse from $V_h = +40\text{ mV}$ to -80 mV (just as there was no ΔF for the pulse back to $+40\text{ mV}$, as shown). This is also true for fig.3e. In this case (fig.3c,e), 100 ms is not sufficient to fully reach steady-state. This is because of the process of conversion. If VSD-I did not convert, the ΔF signals would have been identical across fig.3b-e.

Our intention for these panels (fig.3b-e) was to exemplify an experimental detection of conversion. This is why we used one specific pulse (-80 to $+40\text{ mV}$) that (1) was sufficient to show ΔF in F1 transitions (unconverted VSD) and (2) did not show ΔF for F2 transitions (converted VSD).

We have added to the protocol in fig.3b-e, as you recommended in the Minor comments. Again, thank you for urging us to clarify this figure.

Methods: “Four averages were performed to increase the signal-to-noise ratio of fluorescence signals.” Do these averages have been performed for each voltage command? If 20 depolarizations are applied on the same cell, this will imply a total of 80 traces have been measured. Is there any MTS-TAMRA bleaching during this process? If so, how was it accounted for?

Yes, these averages were performed for each voltage command (technically, a full pulse family was performed, and then repeated for averaging). Some bleaching was observed in the traces, and it was mathematically subtracted as standard VCF practice—the goodness of bleaching subtraction is evident in the shown fluorescence traces. Most of the background fluorescence likely comes from autofluorescence and non-specifically bound fluorophores, which do not contribute to ΔF , but do contribute to bleaching. Some gradual reduction of ΔF due to bleaching was accounted-for in the 4-state model fitting (eq. 14).

Minor comments

Results: “To optically track the movements of individual VSDs under physiologically relevant conditions, we used VCF”. The channel is reconstituted with CACNA2D1 and CACNB-2A or -B3 in a *Xenopus* oocyte. The authors mention later that other accessory subunits and/or PIP2 could regulate the channel. Thus, the statement “physiologically relevant conditions” seems overstated since only the essential components of the channel are present in the oocyte.

Thank you for prompting us to clarify this statement. “Relevant” is the operative word here. Accordingly, we have added: ‘By “physiologically relevant conditions”, we mean an in-cellula study of the human, conducting, $Ca_v2.1$ pore-forming isoform, reconstituted with essential auxiliary subunits.’ (p.2, lines 45-46).

Figure 2: It would be important for the readers if the authors explain why the direction of fluorescence change is opposite for VSD-IV compared to VSD-I or VSD-III while the same dye is used for all VSD tracking.

Yes, we have now added this text in the results section (p.3, lines 23-27):

“As in our VCF investigations of $Ca_v1.1$ and $Ca_v2.2$ channels^{34, 35}, the ΔF signals from VSD-IV had opposite sign to those resolved from VSD-I and VSD-III (fig.2d). A straightforward interpretation is that, when MTS-TAMRA labels VSD-I and VSD-III, it is relatively more quenched in the active state than the resting state; and vice-versa for VSD-IV.”

Figure 2f to 2h: consider presenting the fitting parameters in a table for easier comparison of F1 and F2 activation for each VSD.

We have included new Table 1 with this information.

Figure 3b: Consider including the V_h before returning to -80 mV and then to the test pulse in the depolarization protocol cartoon. That would make the protocol easier to understand without looking at the methods section.

We have amended Fig.3b-e.

Methods: “P/6 subtraction was performed to reduce capacitive transients”. The rearrangement of some VSDs occurred at a relatively low voltage (below -80 mV). At which holding voltage P/6 was run?

Thank you for prompting us to clarify our protocols. P/-6 pulses were run at -100 mV (now added to the Methods, p.9, lines 9-10). This is likely one reason why IQ is difficult to detect. Note also that capacitance compensation would have had the same effect. Capacitive transients (and IQ) were neutralized during our oocyte mounting protocol with holding voltage at -90 mV.

Reviewer #2 (Remarks to the Author):

Thank you for your insightful comments and useful feedback. We have addressed all your concerns below (in blue italics), and we hope our responses are to your satisfaction.

In this study, Kaiqian Wang and collaborators under the lead of Antonios Pantazis studied the structural dynamics of the four voltage-sensing domains (VSD) of CaV2.1 calcium channels and derived information about their specific roles in channel gating and voltage-dependent inactivation (VDI). This study addresses a timely and important scientific problem, relevant for both channel biophysics and basic neurosciences.

To tackle this problem, the authors apply voltage-clamp fluorometry of genetically altered and labeled CaV2.1 channels expressed in oocytes, a highly sophisticated technique that allows to track the voltage-dependence and kinetics of the conformational rearrangements of individual VSDs in response to depolarization. The experiments are of the highest standards, the data are convincing and clearly presented in the figures, tables and supplementary information.

Their recordings reveal distinct voltage-dependent activation of VSDs I, III and IV. Consistent with some structural data finding VSD II locked in the resting state, VSD II did not move in response to depolarization. This finding is backed up by numerous controls excluding the possibility that the observed lack of VSD II movement resulted from a technical problem or was dependent on subunit or lipid interactions.

Most intriguingly, in their voltage-clamp fluorometry experiments VSD I displayed a bi-modal voltage dependence, one of which (F1) closely corresponded to channel opening. Different holding potentials favored one or the other state, indicating that this mode shift of VSD I underlies VDI. Co-expressing beta 3 instead of beta 2a shifted the voltage-dependence of the two modes. The data were fitted well with a four-state model. Analyzing the state transitions allowed the authors further characterization of the molecular mechanisms underlying VDI. In conclusion, the results of the study convincingly demonstrate that VSD I controls channel opening and its conversion into mode 2 results in VDI of CaV2.1.

Furthermore, the data analysis with the four-state model reveal numerous intriguing aspects related to CaV2.1 both in experimental situations and in real life. Some of these deserve further (future) investigations, other´s could be discussed here. For example, relating to Figure 6, in what state are the four VSDs at the resting potential of a typical neuron, how do these sequentially transition on depolarization / channel opening? Or, which states do the four VSDs most likely occupy in preparations used for cryo-EM structure analysis?

Thank you for prompting us to better discuss our results. We have rewritten the Discussion, so we more clearly describe the state of the VSDs in the resting potential and how they respond to depolarization (p.5, lines 23-36).

We agree with cryo-EM structures (so far acquired in the absence of electric field, i.e., at 0 mV), that VSDs I, III and IV are in an active conformation, and that VSD II is “locked down” in a resting conformation. Our work further predicts that VSDs I, III and IV are specifically in the converted (mode 2) active state at 0 mV. We have elaborated on our Discussion with this view (p.6, lines 14-16):

“In the structures, all other VSDs were resolved in an active state; this is in agreement with our work, which predicts that the majority of VSD-I, VSD-III, and VSD-IV would be in a mode-2 active state at $V_h = 0$ mV (fig.6).”

Overall, this is a fine study, ready to be published. A couple of questions could be addressed for clarity purposes:

1. The reason why the authors use the heterologous beta 2a is clear. But why did the authors use beta 3 for comparison, rather than beta 4, which represents the foremost partner of CaV1.2 in CNS neurons?

This is a good question that we grappled with at the start of the project. Since beta 2a inhibits VDI the most, we opted to compare it against beta 3, as it accelerates VDI the most (Stea et al PMID: 7524096; DOI: [10.1073/pnas.91.22.10576](https://doi.org/10.1073/pnas.91.22.10576)).

We have added these lines on our Discussion (p.6, lines 38-42):

“While both β_2 and β_3 subunits are expressed in the brain, a highly abundant $Ca_v\beta$ isoform is β_4 ⁶. Since the initial rate of inactivation in the presence of β_4 is roughly half-way between β_{2a} (slowest) and β_3 (fastest)²⁷, we anticipate that the properties of native $Ca_v2.1$ VSDs are in-between those reported here.”

2. VSD-I interconversion from F1 (available) to F2 (inactivated) is a highly plausible explanation of the bi-modal voltage-dependence of activation. Classical charge interconversion of LTCC is characterized by the constancy of total gating charges moved in the two modes. Can the authors analyze their fluorometry data to demonstrate that the total amount of conformational conversion remains constant at the different holding potentials? In the normalized plots in Fig. 3 this information is lost.

You are correct, and we agree that conversion should remain constant.

This was not the case for the Boltzmann fits in fig.3. However, these fits were empirical: they did not contain conversion transitions, which may produce ΔF , nor a systematic correction for bleaching between families of pulses at different holding potentials.

So, we imposed “ ΔF conservation” in our 4-state model fits, by adding an additional constraint to the state fluorescence levels. Specifically, the fluorescence of state R2 (F_{R2}), previously a free parameter, was now constrained to be $F_{A2} - F_{A1}$ (new eq. 13).

The constraint on F_{R2} works similarly to the “charge conservation” constraint (eq.10), imposing “conformational conservation”. Despite reducing the number of free parameters by 1, the goodness of the fits was practically unchanged (new fig.S3), and some confidence intervals decreased (new table S2). This constraint also supports that all conformational change (ΔF) is accounted for by the four state transitions.

As a result of re-fitting with this constraint, figures 5, 6 and S3, and movie S1, have subtle differences from the originals. Table S2 also has some changes in the fitting parameters, and we added new table S3 (with rate equilibria) to aid in our discussion. Accordingly, the discussion on the effects of beta subunits has been modified to reflect changes in the model fitting parameters (p.6, lines 32-33). The overall conclusions of our paper did not change.

Thank you for prompting us to implement this important consideration.

Minor:

Line 74 For an optical recording technique, the expression, “faint ΔF signal” can be misunderstood to mean that the fluorescence signal itself was faint (rather than the delta), thus implying a technical problem.

You are right, we meant that the delta was faint. Background F was normal. Rephrased to “faint $\Delta F/F$ signal”.

Lines 682-684 The cited references refer to CaV2.2 and 2.3 channels, not to CaV2.1, although the mechanism may be the same. Make clear!

We added “in related Ca_v2 channels”

Suppl. Data:

Line 58 “normalized (a)” should be (c)

Fixed.

Reviewer #3 (Remarks to the Author):

Thank you for your constructive feedback.